# GPR180 is a component of TGFβ signalling that promotes thermogenic adipocyte function and mediates the metabolic effects of the adipocyte-secreted factor CTHRC1

Lucia Balazova[1], Miroslav Balaz[1,2,3], Carla Horvath[1], Áron Horváth[4,5], Caroline Moser[1], Zuzana Kovanicova [2], Adhideb Ghosh [1,6], Umesh Ghoshdastider[1], Vissarion Efthymiou[1], Elke Kiehlmann[1], Wenfei Sun [1], Hua Dong [1], Lianggong Ding[1], Ez-Zoubir Amri[7], Pirjo Nuutila [8], Kirsi A. Virtanen[8], Tarja Niemi[9], Barbara Ukropcova[2,10], Jozef Ukropec[2], Pawel Pelczar[11], Thorsten Lamla[12], Bradford Hamilton[13], Heike Neubauer [13] & Christian Wolfrum [1✉]

Activation of thermogenic brown and beige adipocytes is considered as a strategy to improve metabolic control. Here, we identify GPR180 as a receptor regulating brown and beige adipocyte function and whole-body glucose homeostasis, whose expression in humans is associated with improved metabolic control. We demonstrate that GPR180 is not a GPCR but a component of the TGFβ signalling pathway and regulates the activity of the TGFβ receptor complex through SMAD3 phosphorylation. In addition, using genetic and pharmacological tools, we provide evidence that GPR180 is required to manifest Collagen triple helix repeat containing 1 (CTHRC1) action to regulate brown and beige adipocyte activity and glucose homeostasis. In this work, we show that CTHRC1/GPR180 signalling integrates into the TGFβ signalling as an alternative axis to fine-tune and achieve low-grade activation of the pathway to prevent pathophysiological response while contributing to control of glucose and energy metabolism.

[1] Institute of Food, Nutrition and Health, ETH Zürich, 8603 Schwerzenbach, Switzerland. [2] Institute of Experimental Endocrinology, Biomedical Research Center at the Slovak Academy of Sciences, 84505 Bratislava, Slovakia. [3] Department of Animal Physiology and Ethology, Faculty of Natural Sciences, Comenius University in Bratislava, 84215 Bratislava, Slovakia. [4] Biomechanics Laboratory, University Hospital Balgrist, University of Zurich, 8008 Zurich, Switzerland. [5] Institute of Biomechanics, ETH Zurich, 8093 Zurich, Switzerland. [6] Functional Genomics Centre Zurich, ETH Zurich/ University of Zurich, 8057 Zurich, Switzerland. [7] Université Côte d'Azur, French National Centre for Scientific Research, Inserm, iBV, 06107 Nice, France. [8] Turku PET Centre, University of Turku, 20520 Turku, Finland. [9] Department of Surgery, Turku University Hospital, 20520 Turku, Finland. [10] Institute of Pathophysiology, Faculty of Medicine, Comenius University, 81108 Bratislava, Slovakia. [11] Center for Transgenic Models, University of Basel, 3350 Basel, Switzerland. [12] Drug Discovery Sciences, Boehringer Ingelheim Pharma GmbH & Co. KG, 88397 Biberach an der Riss, Germany. [13] Cardiometabolic Diseases Research Department, Boehringer Ingelheim Pharma GmbH and Co. KG, 88397 Biberach an der Riss, Germany. ✉email: christian-wolfrum@ethz.ch

Obesity and the associated diseases such as type 2 diabetes, dyslipidaemia or cardiovascular complications represent a major health burden. Since obesity is a consequence of chronic positive energy balance, effective strategies to combat the disease must target food intake and/or energy expenditure. In contrast to white adipose tissue (WAT), which stores excessive energy in the form of lipids, brown adipose tissue (BAT) is a highly metabolically active tissue capable of dissipating chemical energy in the form of heat[1]. In response to certain environmental, hormonal and pharmacological stimuli, beige/brite adipocytes that morphologically and functionally resemble brown adipocytes arise in WAT depots[2–4]. The unique capacity of these adipocytes to burn energy is enabled by uncoupling protein 1 (UCP1) present in the inner mitochondrial membrane, which uncouples the proton gradient generated by the respiratory chain from ATP synthesis[5]. Brown and beige adipocytes are dependent on glucose and free fatty acids to cover the high-energy demands of the rich mitochondrial network. Thus, active brown/beige adipocytes serve as a metabolic sink for these substrates and therefore are suggested as a target tissue which can be activated to ameliorate metabolic diseases[6]. Importantly, physiologically relevant amount of brown/beige adipocytes exist in adult humans[7–9] and their appearance is associated with increased energy expenditure, lower adiposity and reduced risk of insulin resistance[10,11], further supporting its role in the control of metabolic homeostasis. So far, the most potent stimulus to activate thermogenesis is noradrenaline released by sympathetic nerve endings upon cold exposure[1]. Pharmacological activation of brown/beige adipocytes by the use of a selective β3-adrenergic receptor agonist mirabegron requires high doses, which is accompanied by adverse side effects such as an increase in blood pressure and heart rate[12,13]. Therefore, identification of alternative pathways to increase thermogenic activity of brown/beige adipocytes is of great interest.

Transforming growth factor β (TGFβ) signalling pathway controls cellular homeostasis in multiple tissues and its aberrant responsiveness is associated with wide range of human pathologies including autoimmune, gastrointestinal and fibrotic diseases, as well as cancer[14,15]. Recently, several studies demonstrated that TGFβ pathway inhibits adipogenesis and formation of brown adipocytes[16–18]. Nevertheless, its function in mature adipocytes is unexplored. Pharmacological targeting of the canonical TGFβ cascade is controversial due to the pleiotropy of TGFβ action. However, it is well accepted that the TGFβ signalling machinery is complex and involves SMAD-independent signalling and many co-receptors including G protein-coupled receptors (GPCRs), which further fine-tune the activity of the pathway[19].

Here, we show that the orphan GPR180 receptor mediates the thermogenic action of Collagen triple helix repeat containing 1 (CTHRC1) protein to control metabolic homeostasis. We propose that CTHRC1/GPR180 signalling integrates with the TGFβ signalling machinery to induce low-grade activation of the pathway and to fine-tune the TGFβ1 response, to maintain the activity of SMAD3 within the hormetic range and preserve the physiological function of the tissue.

## Results

**GPR180 is required for proper beige adipocyte function**. To identify novel membrane receptors with the potential to promote thermogenic adipocyte activity, we compared the transcriptome of human supraclavicular BAT (scBAT) and adjacent subcutaneous WAT of six healthy young volunteers, and cross-analysed it with the transcriptome of human multipotent adipose-derived stem (hMADS) cells differentiated into beige and white adipocytes. The overlap of these two datasets revealed 1012 differentially expressed genes (DEGs) specific for mature adipocytes, which were further filtered for surface receptors (Fig. 1a; Supplementary Data 1). Using this approach, we identified 29 receptors differentially expressed between human BAT and WAT. One of the top hits was GPR180, which is upregulated in brown fat on both cellular and tissue level (Supplementary Fig. 1a and 1b). As GPCRs represent one of the most important integral membrane protein families and serve as attractive drug targets, due to their relevance in the treatment of various diseases, we decided to focus on GPR180 to investigate the role of this orphan receptor in the regulation of thermogenic adipocyte function.

Silencing of the gene encoding GPR180 (Supplementary Fig. 1c) in mature beige hMADS cells led to a reduction of UCP1 expression on both mRNA and protein level (Fig. 1b and Supplementary Fig. 1d), which was associated with attenuated cAMP-stimulated uncoupled respiration to the extent that cells with ablated GPR180 resembled white adipocytes (Fig. 1c). Consistently, several brown and beige adipocyte markers were decreased following GPR180 knockdown, while the expression of general adipocyte markers was not affected (Supplementary Fig. 1d). We found that lack of GPR180 signalling affected specifically UCP1 content, independent of any changes in the percentage of mature adipocytes (Supplementary Fig. 1e and f) or expression of mitochondrial protein complexes mediating oxidative phosphorylation (Supplementary Fig. 1g). Suppression of the brown adipocyte phenotype following Gpr180 knockdown was observed also in immortalized murine brown adipocytes (Supplementary Fig. 1h, i and j). Contrary, overexpression of GPR180 in mature human white adipocytes (Fig. 1d) resulted in an increase in UCP1 protein level (Fig. 1e), as well as in higher cAMP-stimulated uncoupled and maximal mitochondrial respiration (Fig. 1f). A detailed quantification demonstrated that knockdown of GPR180 in preadipocytes did not affect the process of adipogenesis (Supplementary Fig. 1k and l) in line with its low expression in undifferentiated preadipocytes (Supplementary Fig. 1m). These data suggest that GPR180 is essential for thermogenic function of mature brown and beige adipocytes and its ablation shifts brown and beige adipocytes towards a white-like phenotype.

**Metabolic derangements in *Gpr180* knockout mice**. To study the physiological relevance of GPR180 signalling, we generated a *Gpr180* global knockout mouse by the CRISPR/Cas9 technology (Supplementary Fig. 2a). In accordance with our previous observations, UCP1 expression was downregulated in both interscapular BAT (iBAT) (Fig. 2a and Supplementary Fig. 2a) and inguinal WAT (iWAT) (Fig. 2b) of *Gpr180* ablated mice, although we did not observe any difference in body weight (Supplementary Fig. 2b) or body composition (Supplementary Fig. 2c and d) when animals were fed with a chow diet. Concomitant with a decrease in UCP1 protein in both iBAT and iWAT, a significant reduction in energy expenditure for several timepoints with an overall strong trend related to genotype ($p = 0.08$) was observed in *Gpr180* knockout mice (Fig. 2c), while the locomotor activity (Supplementary Fig. 2e), food intake (Supplementary Fig. 2f), as well as substrate utilization (Supplementary Fig. 2g) remained unchanged. In line with these results, we observed a significantly reduced surface temperature in *Gpr180* knockout mice in response to administration of a selective β3-adrenoreceptor agonist, suggesting impaired brown adipocyte thermogenic activity (Fig. 2d and e). Importantly, mice lacking *Gpr180* showed elevated fasting blood glucose levels (Supplementary Fig. 2h) and displayed impaired glucose tolerance (Fig. 2f), although fasting insulin levels were not altered (Supplementary Fig. 2i). When challenged with high-fat diet

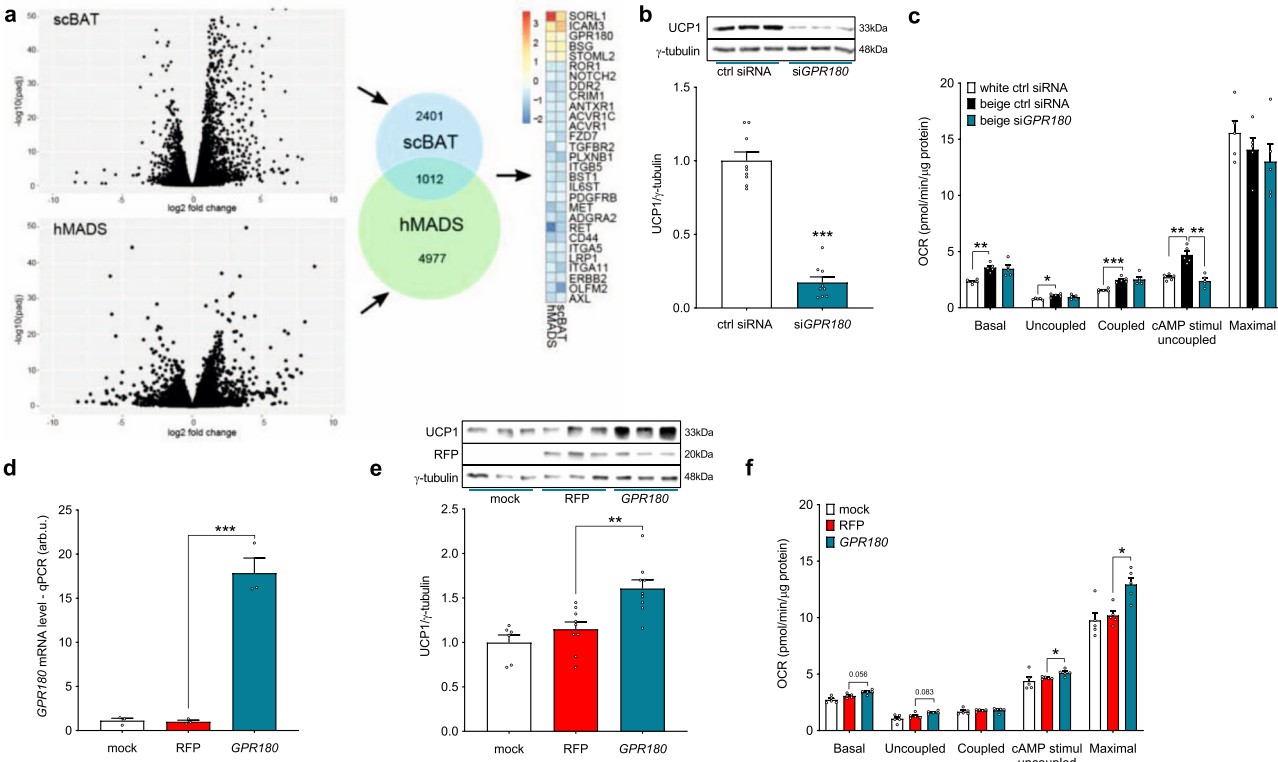

**Fig. 1 GPR180 is upregulated in BAT and required for the brown phenotype. a** Workflow of target identification by transcriptomic analysis of human supraclavicular BAT, subcutaneous WAT and hMADS cells differentiated into beige and white adipocytes. DESeq2 detects DE genes based on a generalized linear model using the negative binomial distribution. Effect of *GPR180* silencing in human beige adipocytes on **b** UCP1 protein (*n* = 9; *p* < 0.0001) and **c** mitochondrial respiration (*n* = 5; *p* = 0.0016 for cAMP-stimulated uncoupled respiration). **d** Quantification of lentiviral *GPR180* overexpression on mRNA level in white adipocytes (*n* = 3; *p* < 0.0001) and its effect on **e** UCP1 protein (*n* = 9; *p* = 0.0021) and **f** mitochondrial oxygen consumption (*n* = 5; *p* = 0.0437 for cAMP-stimulated uncoupled respiration and *p* = 0.0161 for maximal respiration). Data are presented as mean ± SEM. Statistical analysis was performed by two-sided Student´s *t*-test (**b**), one-way ANOVA with Dunnett's post-hoc test (**d**, **e**) or two-way ANOVA with Tukey's post-hoc test (**c, f**). Significance is indicated as **p* < 0.05, ***p* < 0.01 and ****p* < 0.001. cAMP cyclic adenosine monophosphate, GPR180 G protein-coupled receptor 180, hMADS human multipotent adipose-derived stem cells, OCR oxygen consumption rate, RFP red fluorescent protein, scBAT supraclavicular brown adipose tissue, UCP1 Uncoupling protein 1.

(HFD), *Gpr180* knockout mice gained more weight (Fig. 2g) and this was accompanied by higher fat mass accumulation (Supplementary Fig. 2j and k), pronounced liver steatosis and increased plasma ALT activity indicating an early stage of liver disease (Fig. 2h, Supplementary Fig. 2l and m). Based on the results of the molecular analyses of adipose tissues and the metabolic phenotyping of the global knockout mice we assumed that dysfunctional iBAT and/or impaired adipose tissue browning might be responsible for the accelerated development of metabolic disturbances in *Gpr180* knockout mice. To test this hypothesis, we housed animals at thermoneutrality (30 °C, chow diet) for 8 weeks, followed by a glucose tolerance test and a 12-week HFD feeding regime. In this experimental setting, *Gpr180* knockout mice did not exhibit any difference in glucose tolerance (Fig. 2i), body weight gain (Fig. 2j) or hepatic lipid accumulation (Fig. 2k) indicating that the adverse metabolic phenotype in this mouse model is most likely mediated by decreased thermogenic activity of brown and beige fat. To confirm our findings, we generated an inducible adipocyte-specific *Gpr180* (*Gpr180^{fl/fl}* × *Adip-CreERT2*) knockout mouse (Supplementary Fig. 2n). In agreement with our previous observations, ablation of *Gpr180* specifically in mature adipocytes resulted in reduced UCP1 protein levels in iWAT (Fig. 2l and m), reduced energy expenditure (Fig. 2n) and surface temperature (Fig. 2o and p). This effect was not due to changes in locomotor activity or food intake (Supplementary Fig. 2o and p). Interestingly, mice with

adipocyte-specific ablation of *Gpr180* utilized less lipids as indicated by a shift in the respiratory quotient (Supplementary Fig. 2q), which was supported by significantly reduced levels of random fed blood glucose and increased free fatty acid levels (Supplementary Fig. 2r and s). Furthermore, adipocyte-specific *Gpr180* knockout mice developed impaired glucose tolerance (Fig. 2q) and showed significantly higher body weight gain when challenged with HFD (Fig. 2r). Altogether, we show here that loss of GPR180 receptor in adipose tissue diminishes both brown and beige adipocyte function, which leads to impaired metabolic control.

**GPR180 is not a GPCR but a component of the TGFβ signalling pathway.** To study the mechanism of GPR180 action and the downstream signalling cascade, we performed a pathway analysis of DEGs obtained by RNA-seq of beige hMADS cells following *GPR180* silencing. Surprisingly, there was no evidence for GPCR mediated signalling (Supplementary Fig. 3a and Supplementary Data 2). In addition, knockdown of *GPR180* had no effect on cAMP levels, phosphorylation of PKA substrates, IP1 and calcium levels in mature beige adipocytes (Supplementary Fig. 3b–f). However, results of the above-mentioned analysis suggest an involvement of GPR180 in TGFβ signalling (Fig. 3a). As TGFβ signalling is mediated by SMAD3, we analysed SMAD3 phosphorylation in response to *GPR180* knockdown. In line with

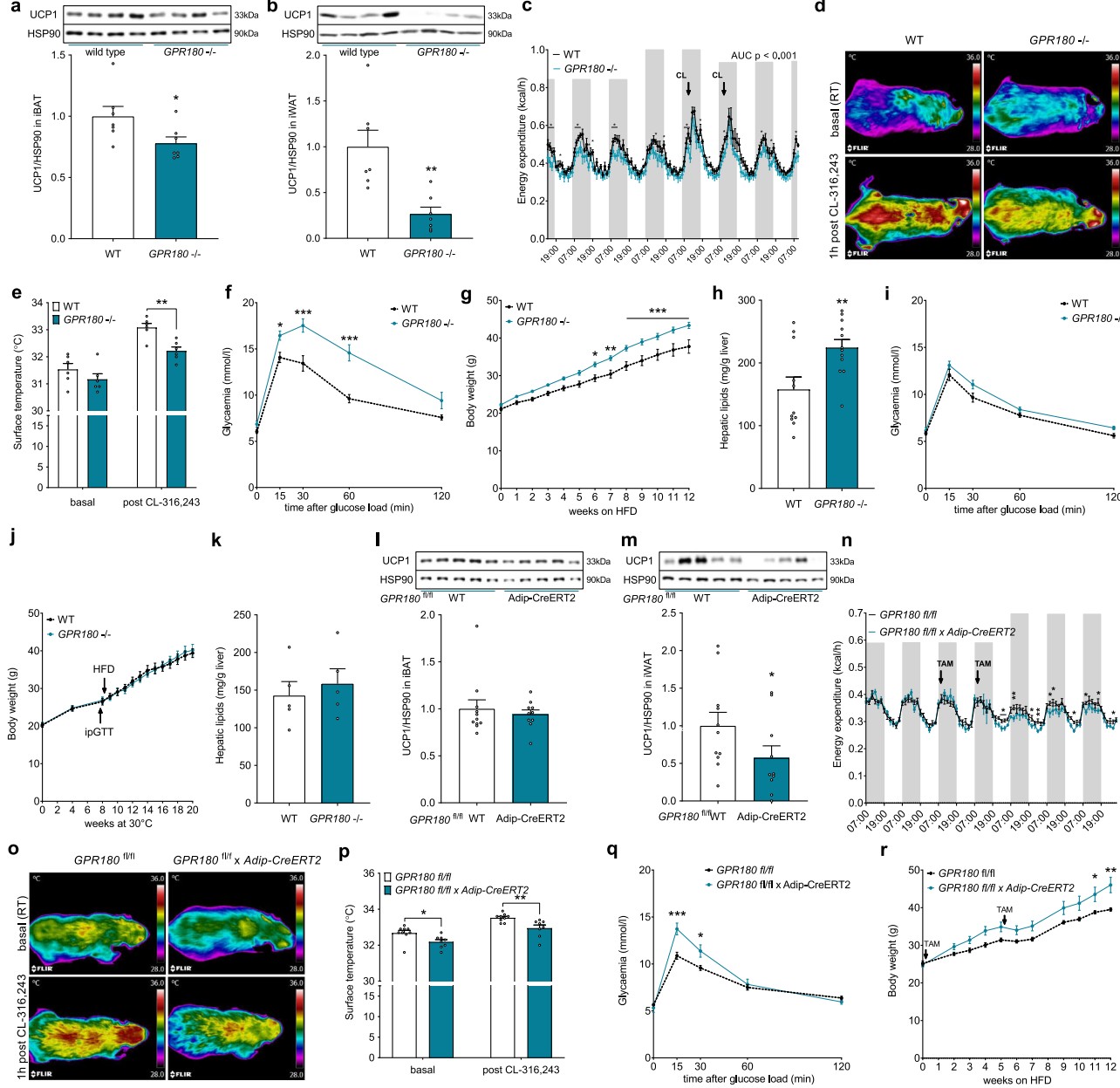

the RNA-seq data, we observed a significant reduction in phosphorylation of SMAD3 protein at serine 423 upon *GPR180* ablation in beige adipocytes (Fig. 3b). We could not confirm the regulation of other pathways predicted by the DEG analysis, such as MAPK signalling, nor did we observe any difference in ERK or P38MAPK phosphorylation (Supplementary Fig. 3g and h). To validate the role of GPR180 in TGFβ signalling, we tested whether the activity of TGFβ1, the key regulator of SMAD3 phosphorylation, is dependent on presence of GPR180. Interestingly, TGFβ1-induced phosphorylation of SMAD3 was attenuated in beige adipocytes in absence of GPR180 (Fig. 3c). Similar results were obtained also in HEK-293T cells (Supplementary Fig. 3i). On the other hand, overexpression of GPR180 in white adipocytes increased basal SMAD3 phosphorylation (Fig. 3d), which indicates that GPR180 is required to achieve full TGFβ signalling. Based on this data and the fact that we could not find any changes in G protein-coupled signalling (Supplementary Fig. 3b–f), we hypothesized that GPR180 is not a GPCR and therefore its topology might diverge from the one characteristic for canonical GPCRs. To investigate the GPR180 orientation in plasma membrane, we overexpressed GPR180 fused with either a V5 tag at carboxy terminus or a HA tag at amino terminus in hMADS cells. We detected C-terminal V5 tag in both permeabilized and non-permeabilized conditions suggesting its extracellular localization (Fig. 3e and f). In contrast, HA tag could be detected only if the adipocytes were permeabilized indicating intracellular localization of the N-terminus. Taken together, our data demonstrate that GPR180 is not a GPCR but rather a component of the TGFβ signalling pathway.

**TGFβ enhances mature beige adipocyte function**. Since we could show that GPR180 is a component of the TGFβ signalling pathway, we next investigated the expression profile of other receptors of this pathway in mature adipocytes. We found that both white and beige human adipocytes express all three TGFβ receptors and while *TGFβR1* and *TGFβR2* are abundantly expressed during the whole differentiation, mRNA levels of *TGFβR3* are transiently induced after induction of adipogenic differentiation (Supplementary Fig. 3j). Although recent studies

**Fig. 2 Metabolic derangements in *GPR180* knockout mice are caused by dysfunctional BAT.** UCP1 protein in **a** iBAT ($p = 0.0425$) and **b** iWAT ($p = 0.0027$) of *GPR180* global knockout mice and their wild-type littermates ($n = 7$) fed chow diet and housed at room temperature (RT). **c** Energy expenditure in male mice with deleted *GPR180* ($n = 6$) on chow diet and housed at RT (AUC $p < 0.0001$). **d** Representative images including rainbow scale bar indicating temperature range with min 28 °C and max 36 °C and **e** quantification of surface temperature in male *GPR180* global knockout mice and wild-type littermates ($n = 6$; $p = 0.0065$). **f** Intraperitoneal glucose tolerance test in 12-weeks-old male mice housed at RT and fed chow diet (WT $n = 6$, $GPR180^{-/-}$ $n = 7$; $p = 0.047$ at 15 min, $p = 0.0001$ at 30 min and $p < 0.0001$ at 60 min). **g** Body weight gain ($p = 0.0212$ at 6 weeks, $p = 0.0036$ at 7 weeks, $p = 0.0006$ at 8 weeks, $p = 0.0003$ at 9 weeks, $p = 0.0004$ at 10 weeks, $p < 0.0001$ at 11 and 12 weeks), and **h** hepatic lipid accumulation ($p = 0.008856$) in male mice housed RT and fed HFD for 12 weeks (WT $n = 11$, $GPR180^{-/-}$ $n = 12$). **i** Glucose tolerance test in 12-weeks-old male mice housed at thermoneutrality (TN) for 8 weeks prior the test and fed chow diet ($n = 5$). **j** Body weight gain and **k** hepatic lipid accumulation in animals housed at TN and fed HFD ($n = 5$). Representative blots and quantification of the UCP1 protein levels in **l** iBAT and **m** iWAT ($p = 0.0473$) of male adipocyte-specific *GPR180* (aGPR180) knockout mice and fl/fl controls (fl/fl control $n = 11$, aGPR180 knockout $n = 10$). **n** Energy expenditure in male aGPR180 knockout mice and fl/fl controls (fl/fl control $n = 6$, aGPR180 knockout $n = 5$; $p = 0.0473$; 0.0051; 0.0450; 0.0098; 0.0139; 0.0412; 0.0277; 0.0212; 0.0278). **o** Representative images including rainbow scale bar indicating temperature range with min 28 °C and max 36 °C and **p** quantification of surface temperature ($p = 0.0255$ for basal and $p = 0.0086$ for post CL-316,423) in male aGPR180 knockout mice and fl/fl controls (fl/fl control $n = 9$, aGPR180 $n = 8$). **q** Glucose tolerance test in male aGPR180 knockouts and fl/fl controls ($n = 4$; $p = 0.0002$ at 15 min and $p = 0.0243$ at 30 min) 2 weeks after tamoxifen (TAM) gavage (2 mg/animal) in two consecutive days while mice were housed at RT and fed chow diet. **r** Body weight gain in aGPR180 knockout mice fed HFD housed at RT (fl/fl control $n = 6$, aGPR180 knockout $n = 7$; $p = 0.0419$ at 11 weeks and $p = 0.0011$ at 12 weeks). Data are presented as mean ± SEM. Statistical significance was calculated using two-sided (**a–c**, **h**, **k**) and one-sided Student´s *t*-test (**l**, **m**) or two-way ANOVA with repeated measurements followed by Sidak post-hoc test (**e–g**, **i**, **j**, **p–r**) and Fisher' LSD multiple comparison test (**n**). Area under the curve was calculated to compare energy expenditure in the global knockout mice (**c**). Statistical differences are indicated as *$p < 0.05$, **$p < 0.01$ and ***$p < 0.001$. AUC area under the curve, CL CL-316,243, GPR180, G protein-coupled receptor 180, HFD high-fat diet, HSP90 Heat shock protein 90, iBAT interscapular brown adipose tissue, iWAT inguinal white adipose tissue, RT room temperature, TAM tamoxifen, UCP1 Uncoupling protein 1, WT wild-type.

extensively explored the effect of TGFβ/SMAD signalling cascade on adipocyte formation[16–18], experimental evidence on its action in mature adipocytes is missing. Interestingly, treatment of hMADS cells-derived mature beige adipocytes with TGFβ1 dose-dependently increased SMAD3 phosphorylation (Supplementary Fig. 3k), which resulted in UCP1 upregulation (Fig. 3g) along with a marked increase in cAMP-stimulated uncoupled and maximal mitochondrial respiration (Fig. 3h), suggesting that the function of TGFβ signalling in regulation of mature beige adipocyte activity and beige adipogenesis might be different. We therefore investigated how manipulation of the individual components of TGFβ pathway affects beige adipocyte function. Pharmacological blockade of TGFβ pathway with SB-431542, a selective TGFβR1 kinase inhibitor, led to reduced SMAD3 phosphorylation (Supplementary Fig. 3l) and UCP1 protein expression (Fig. 3i), while mitochondrial respiration was not altered (Supplementary Fig. 3m). In addition, genetic inhibition of the TGFβ pathway with the aid of siRNAs targeting either TGFβR1, a receptor kinase responsible for SMAD3 phosphorylation, or the downstream signal transducers SMAD3 and SMAD2, downregulated UCP1 protein level and mitochondrial respiration in human hMADS cells (Fig. 3j–o and Supplementary Fig. 3n–p). Silencing of TGFβR2, the ligand-binding receptor, decreased UCP1 expression (Fig. 3k), but did not affect the mitochondrial oxygen consumption rate in human hMADS cells (Fig. 3m), which might be due to a compensation by increased lipolysis (Fig. 3l), as high free fatty acid levels can directly uncouple the inner mitochondrial membrane[20]. Based on these data, we conclude that inhibition of the TGFβ signalling pathway at the level of receptors, or the post-receptor signalling cascade using pharmacological and genetic means supresses adipocyte thermogenic function, thus emphasizing necessity of TGFβ signalling activation to promote the brown/beige adipocyte phenotype.

Since we could show that GPR180 might be another component of TGFβ signalling pathway, we studied the role of GPR180 in chronic TGFβ signalling. Unlike the effect of GPR180 loss on acute TGFβ signalling which was reduced by approximately 25% (Fig. 3c), long-term TGFβ1 treatment (72 h) increased SMAD3 phosphorylation and UCP1 protein levels in beige adipocytes even in absence of GPR180 (Supplementary Fig. 3q and r), suggesting that partial activation of the TGFβ machinery in the absence of GPR180 is sufficient to induce browning. Taken together, we conclude that GPR180 is required for full activation of the TGFβ signalling machinery, however even submaximal activation is sufficient to enhance adipocyte browning.

**GPR180 transduces signal of adipokine CTHRC1 to induce SMAD3.** Several arguments speak for the existence of a novel endogenously produced ligand activating GPR180 in human adipocytes in auto/paracrine manner. First, we observed basal SMAD3 phosphorylation in unstimulated conditions in hMADS cells that are cultured in serum-free medium; next, genetic manipulation of GPR180 levels affected SMAD3 phosphorylation directly. To investigate contribution of endogenous TGFβ secretion to SMAD3 phosphorylation, we quantified TGFβ levels in hMADS cells-conditioned media using a highly sensitive ELISA kit. We did not observe any secretion of TGFβ isoforms by mature human adipocytes into media (Fig. 4a and b, Supplementary Data 3), pointing to species differences in TGFβ secretion by preadipocytes and adipocytes between human and mouse[21]. In addition, we observed a persistent increase of SMAD3 phosphorylation in GPR180 overexpressing adipocytes, which were treated with TGFβ neutralizing antibody (Fig. 4c), further supporting this hypothesis. To identify potential ligands of GPR180, we first analysed the transcriptome of hMADS cells differentiated into mature adipocytes. We identified 315 genes that are significantly expressed in human mature adipocytes and encode for a secreted protein, as evidenced by the presence of a signal peptide sequence (Supplementary Data 4). In addition, proteomic analysis of cell-conditioned media using liquid chromatography coupled to tandem mass spectrometry revealed 322 proteins secreted into media in relevant amounts (Supplementary Data 3). By cross-analysis of transcriptomic and proteomic datasets, we could identify 82 secreted molecules. After exclusion of proteins annotated as extracellular matrix components or involved in regulation of immune response, we obtained 36 potential ligands (Fig. 4a). Since we showed that ablation of GPR180 leads to downregulation of UCP1 protein levels, we performed a screen aimed at identification of secreted proteins whose knockdown will decrease UCP1 levels. Using this

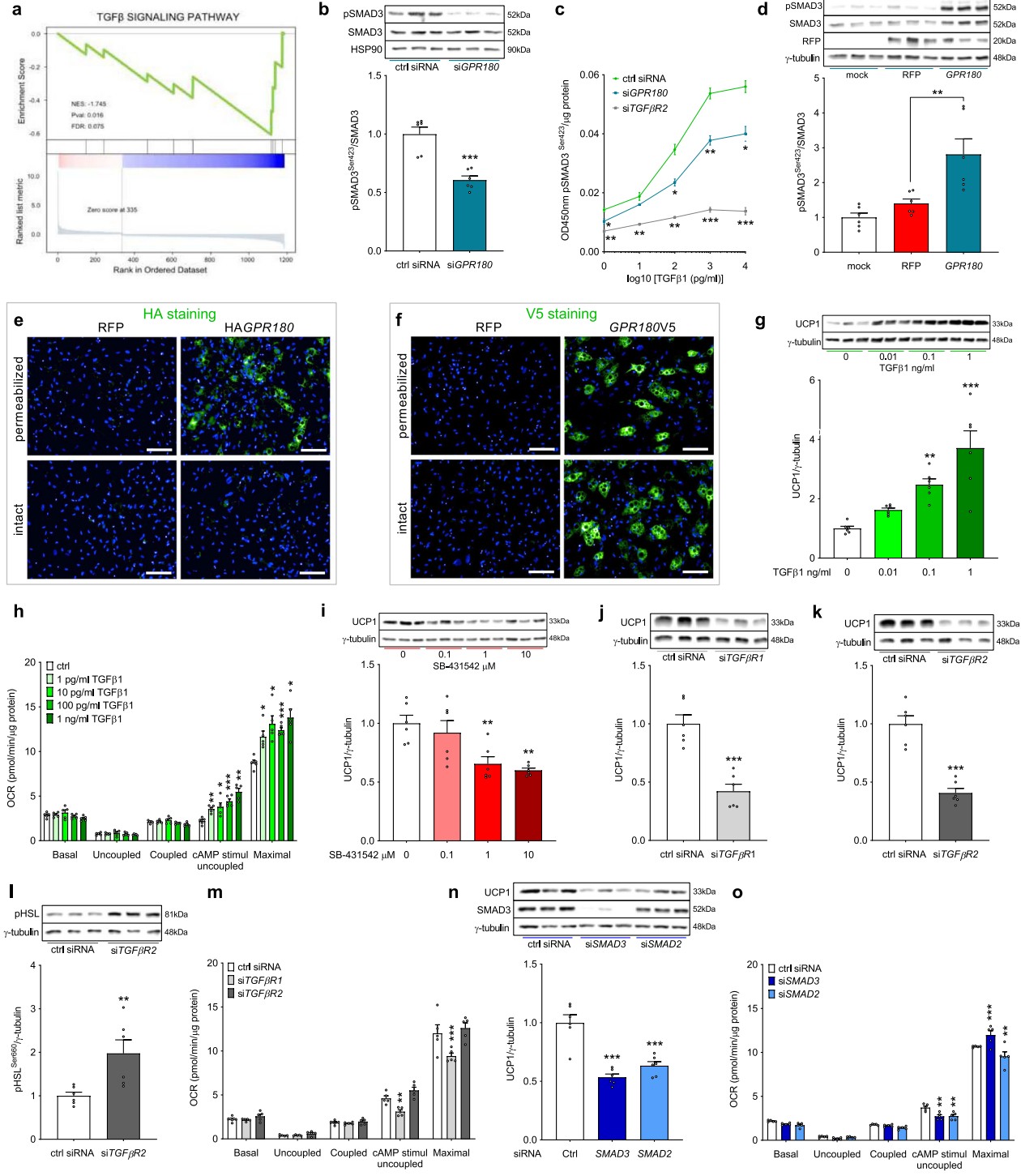

approach, we found that silencing of *CTHRC1*, *FSTL1* and *IGFBP7* reduced UCP1 protein content (Supplementary Fig. 4a), however only CTHRC1 and FSTL1 increased phosphorylation of SMAD3 in beige adipocytes, (Supplementary Fig. 4b). While the stimulatory effect of FSTL1 on SMAD3 phosphorylation was independent on GPR180 (Supplementary Fig. 4c), the effect of CTHRC1 could be completely attenuated by knockdown of this receptor (Fig. 4d), suggesting that CTHRC1 is a regulator of GPR180 function. We could confirm that CTHRC1 protein promotes SMAD3 phosphorylation in beige hMADS cells in a time-dependent manner (Fig. 4e) and that this effect was not due to TGFβ secretion or contamination, utilizing a neutralizing

TGFβ antibody (Fig. 4f). Furthermore, CTHRC1 increased percentage of phosphorylated SMAD3 positive nuclei (Fig. 4g and h), and at the same time, immunostaining of total SMAD3 was increased in the nucleus following CTHRC1 treatment (Fig. 4g and i) indicating translocation of SMAD3 from the cytosol into the nucleus and thereby functional and transcriptionally active SMAD3 signalling. To demonstrate a direct interaction between cell surface GPR180 and CTHRC1, we generated a stable *GPR180* knockout HEK-293T cell line using CRISPR/Cas9 and a construct to overexpress CTHRC1 fused with a HiBiT tag at C terminus, which is secreted into media and thus can be used for binding studies. Using this approach, we could demonstrate that

**Fig. 3 GPR180 is not a GPCR, but a component of TGFβ signalling pathway enhancing mature beige adipocytes function. a** KEGG pathway analysis of differentially expressed genes (DEGs) of human beige adipocytes with ablated *GPR180*. Phosphorylation of SMAD3 at serine 423 in **b** non-starved human beige adipocytes after *GPR180* silencing ($n = 6$; $p = 0.0002$), **c** beige adipocytes treated with different concentrations of TGFβ1 in combination with knockdown of *GPR180* or *TGFβR2* (ctrl siRNA vs si*GPR180* $p = 0.0416$ for 1 pg/ml TGFβ1, $p = 0.0129$ for 100 pg/ml TGFβ1, $p = 0.0034$ for 1 ng/ml TGFβ1 and $p = 0.0146$ for 10 ng/ml TGFβ1) and **d** in white adipocytes overexpressing *GPR180* ($n = 6$; $p = 0.0050$). Representative images of epitope tag immunostaining (green), nuclei stained by Hoechst (blue), **e** N-terminal HA tag and **f** C-terminal V5 tag in hMADS cells overexpressing modified GPR180; scale bar 100 μm. Experiment was performed 3 times with similar results. Long-term TGFβ1 treatment (72 h) dose-dependently promotes **g** UCP1 protein ($n = 6$; $p = 0.0084$ for 0.1 ng/ml and $p < 0.0001$ for 1 ng/ml) and **h** mitochondrial respiration ($n = 5$; cAMP uncoupled respiration $p = 0.0020$ for 1 pg/ml TGFβ1, $p = 0.0334$ for 10 pg/ml, $p = 0.0008$ for 100 pg/ml and $p = 0.0012$ for 1 ng/ml; maximal respiration $p = 0.0205$ for 1 pg/ml TGFβ1, $p = 0.0178$ for 10 pg/ml, $p < 0.0001$ for 100 pg/ml and $p = 0.0127$ for 1 ng/ml TGFβ1) in mature human beige adipocytes. Effect of **i** pharmacological (p = 0.0062 for 1 μM and $p = 0.0017$ for 10 μM) and **j** genetic ($p = 0.0002$) inhibition of TGFβR1 on UCP1 protein level in beige hMADS cells ($n = 6$). Effect of *TGFβR2* silencing on **k** UCP1 expression ($n = 6$; $p < 0.0001$) and **l** HSL phosphorylation at serine 660 ($n = 6$; $p = 0.0126$). **m** Mitochondrial oxygen consumption rate following knockdown of individual TGFβ receptors in mature beige adipocytes ($n = 5$; $p = 0.0082$ for cAMP-stimulated uncoupled respiration and $p < 0.0001$ for maximal respiration). Effect of SMAD2 ($p = 0.0001$) and SMAD3 ($p < 0.0001$) knockdown on **n** UCP1 protein level ($n = 6$) and **o** mitochondrial respiration ($n = 5$; $p = 0.0039$ for SMAD3 and $p = 0.0044$ for SMAD2 cAMP-stimulated uncoupled respiration; $p = 0.0001$ for SMAD3 and $p = 0.0011$ for SMAD2 maximal respiration) in beige hMADS cells. Data are shown as average ±SEM. Statistical analysis was performed using two-sided Student's *t*-test (**b**, **j**–**l**), one-way ANOVA with Dunnett's post-hoc test (**c**, **d**, **g**, **i**, **n**) or two-way ANOVA with Tukey post-hoc test (**h**, **m**, **o**) and significance is indicated as *$p < 0.05$, **$p < 0.01$ and ***$p < 0.001$. cAMP cyclic adenosine monophosphate, GPR180 G protein-coupled receptor 180, HSL Hormone sensitive lipase, HSP90 Heat shock protein 90, OCR oxygen consumption rate, RFP red fluorescent protein, SMAD3 Mothers against decapentaplegic homolog 3, TGFβ1 Transforming growth factor β1, TGFβR1 Transforming growth factor β receptor type 1, TGFβR2 Transforming growth factor β receptor type 2, UCP1 Uncoupling protein 1.

CTHRC1 binding requires the presence of GPR180, as it was strongly reduced in *GPR180* knockout HEK-293T cells. These data indicate that CTHRC1 might be a ligand of GPR180, suggesting interaction of these two components of TGFβ signalling pathway (Fig. 4j).

As we had demonstrated that GPR180 is part of the TGFβ signalling machinery, we next examined the dependency of CTHRC1-induced SMAD3 phosphorylation on the presence of TGFβ receptors. Pharmacological (Fig. 4k) or siRNA-mediated (Fig. 4l) inhibition of TGFβR1 fully abolished phosphorylation of SMAD3 at serine 423 by CTHRC1. Similarly, silencing of TGFβR2 could almost completely reverse the effect of CTHRC1 on SMAD3 phosphorylation (Fig. 4m). Since CTHRC1 was shown to stimulate different pathways in various cell lines[22–25], we also studied its effect on Gs (Supplementary Fig. 4d-f), Gi (Supplementary Fig. 4f-i) and Gq (Supplementary Fig. 4j and k) signalling as well as other kinases (Supplementary Fig. 4l and m) and SMADs (Supplementary Fig. 4n), however, we did not observe any evidence which would support a role of CTHRC1 in G-protein signalling. These results are in agreement with demonstrated inverted membrane topology of GPR180 receptor. Based on these data, we propose that CTHRC1 activates SMAD3 and together with GPR180 represent components of the TGFβ signalling machinery. This mechanism of SMAD3 regulation is not restricted to adipocytes, as increased phosphorylation of SMAD3 (Supplementary Fig. 4o) in response to CTHRC1 was evident also in HEK-293T cells, and dependent on TGFβR1 kinase (Supplementary Fig. 4p). Similarly, SMAD3 activated by CTHRC1 translocated to the nucleus as shown by increased luciferase activity reflecting higher responsiveness of SMAD binding elements (Fig. 4n) in HEK-293T cell line and, importantly, also this effect could be prevented by genetic deletion of *GPR180*. Surprisingly, we found that co-treatment of human beige adipocytes by CTHRC1 blunted the TGFβ1-induced phosphorylation of SMAD3 (Fig. 4o and p). When we compared the stimulatory potential of CTHRC1 and TGFβ1, we found that the maximal level of SMAD3 phosphorylation induced by 20 nM CTHRC1 corresponds to a low-level stimulation with 40 fM TGFβ1 (Fig. 4q). Moreover, we observed increased CTHRC1 expression and secretion in response to long-term TGFβ1 treatment in beige hMADS cells (Fig. 4r) suggesting the existence of a feedback loop to modulate the TGFβ1 response. Taken

together, we identified CTHRC1 as an adipokine that signals via GPR180 in human mature adipocytes to increase SMAD3 phosphorylation. Importantly, this alternative CTHRC1/GPR180 axis of TGFβ pathway represents a feedback system to fine-tune TGFβ1 response and mediate low-grade activation of SMAD3 signalling.

**CTHRC1 enhances beige adipocyte functionality via GPR180.** In accordance with the secretome screening data, we confirmed expression of CTHRC1 on protein level and its secretion into media by both white and brown human adipocytes (Fig. 5a–c). Although CTHRC1 is more abundantly expressed and secreted by white hMADS cells, it is not differentially expressed between supraclavicular BAT and subcutaneous WAT of human volunteers (Fig. 5d). Therefore, we next investigated the effect of CTHRC1 on beige adipocyte metabolism. Long-term CTHRC1 treatment (72 h) increased UCP1 protein levels (Fig. 5e) and cAMP-stimulated uncoupled as well as maximal mitochondrial oxygen consumption (Fig. 5f), and this effect was evident also in presence of a neutralizing TGFβ antibody (Fig. 5g), ruling out the involvement of endogenous TGFβ or recombinant protein contamination. Importantly, the stimulatory action of CTHRC1 on UCP1 levels (Fig. 5h) and mitochondrial respiration (Fig. 5i) in beige hMADS cells required the presence of GPR180. On the other hand, siRNA-mediated *CTHRC1* silencing in mature beige adipocytes downregulated UCP1 protein (Fig. 5j) along with a reduction of basal and cAMP-stimulated uncoupled respiration (Fig. 5k). In addition, we could show that unlike TGFβ1[16–18], CTHRC1 does not affect adipogenesis and beige adipocyte formation (Supplementary Fig. 5a–c). Altogether, these data indicate that GPR180 is indispensable for CTHRC1-induced stimulation of beige adipocyte functionality.

**GPR180 mediates the beneficial effect of CTHRC1 on energy metabolism.** To test the physiological relevance of CTHRC1 in vivo, we generated an AAV8 expressing *CTHRC1* under the control of a liver-specific LP-1 promoter to increase circulating levels of this factor. As expected, we found elevated CTHRC1 protein in plasma of mice injected with AAV-*Cthrc1* compared to AAV-stuffer (Fig. 6a). Importantly, CTHRC1 overexpression in male C57BL/6 N mice attenuated the HFD-induced body weight

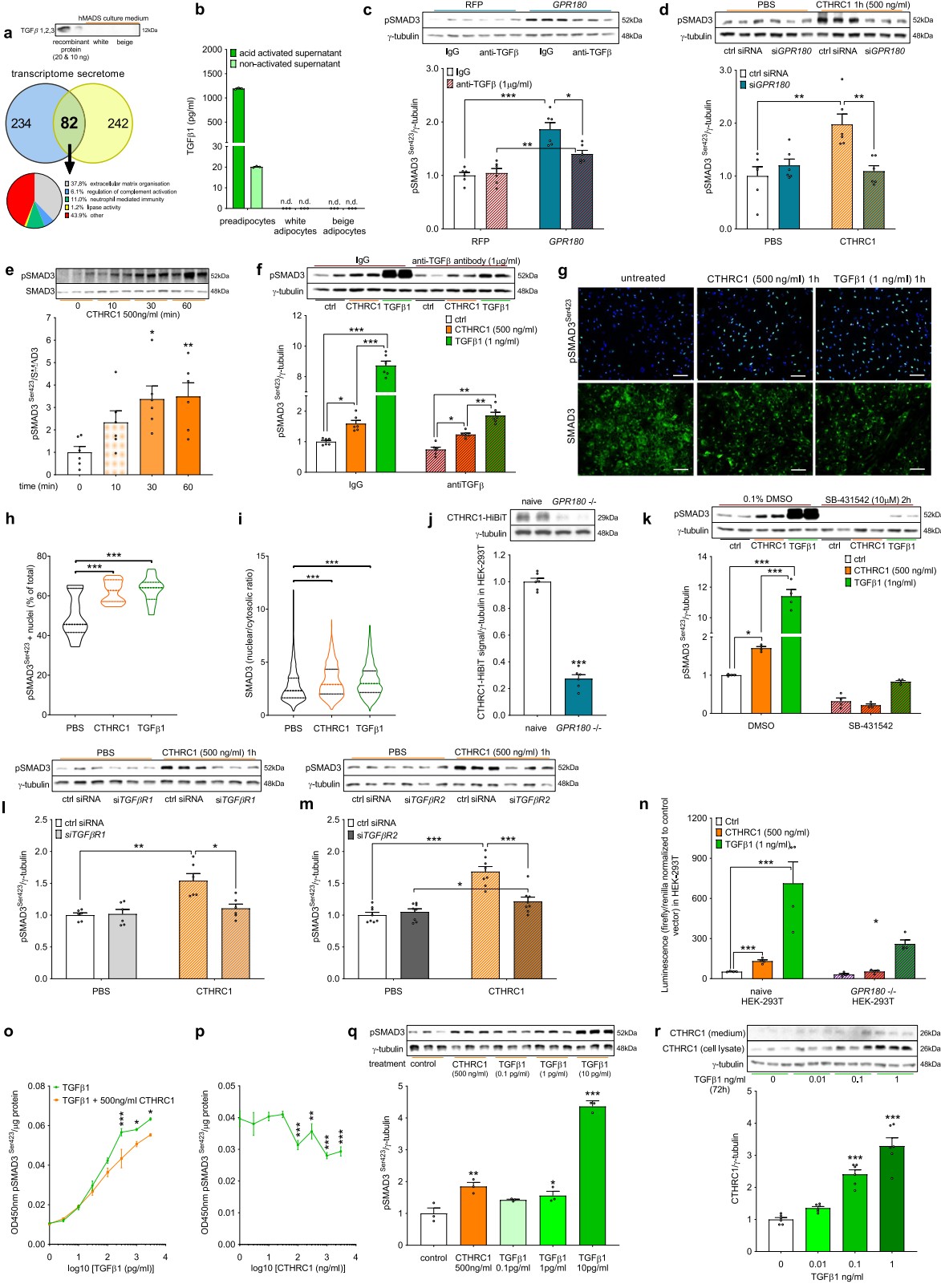

gain (Fig. 6a and Supplementary Fig. 6a), which was mainly due to reduced accumulation of WAT mass (Supplementary Fig. 6b). CTHRC1 overexpression lowered the fasting glycaemia (Fig. 6b) without altering plasma insulin (Supplementary Fig. 6c) or lipid profile (Supplementary Fig. 6d and e), although non-esterified fatty acids levels tended to be elevated (Fig. 6c; $p = 0.09$). A trend towards an increase in phosphorylation of HSL in both iBAT

(Fig. 6d) and iWAT (Fig. 6e) depots suggests that higher fatty acid flux in adipose depots might account for increased tissue activity and contribute to lower body weight gain in CTHRC1 treated mice. This is further supported by upregulated expression of brown adipocyte markers in iWAT (Fig. 6f), while these effects were less prominent in iBAT (Fig. 6g and h). Consistent with these findings, CTHRC1 overexpression in HFD-fed mice led to

**Fig. 4 CTHRC1 induces SMAD3 phosphorylation via GPR180. a** Immunostaining of TGFβ proteins in conditioned media of white and beige hMADS adipocytes with recombinant TGFβ as a positive control and identification of secreted proteins by hMADS adipocytes. **b** TGFβ1 levels in conditioned media of preadipocytes and mature white and beige hMADS adipocytes ($n = 3$). **c** Representative blots and quantification of SMAD3 phosphorylation in white adipocytes overexpressing *GPR180* while treated with TGFβ neutralizing antibody ($n = 6$; $p < 0.0001$ for RFP IgG vs GPR180 IgG; $p = 0.0064$ for GPR180 IgG vs GPR180 anti-TGFβ and $p = 0.0408$ for RFP anti-TGFβ vs GPR180 anti-TGFβ). **d** SMAD3 phosphorylation in beige adipocytes following CTHRC1 treatment in combination with *GPR180* ablation ($n = 6$; $p = 0.0012$ for PBS ctrl siRNA vs CTHRC1 ctrl siRNA and $p = 0.0030$ for CTHRC1 ctrl siRNA vs CTHRC1 si*GPR180*). **e** Time-dependent effect of acute CTHRC1 stimulus on SMAD3 phosphorylation after the adipocytes were starved for 2 h ($n = 6$; $p = 0.0103$ for 30 min and $p = 0.0073$ for 60 min). **f** Effect of CTHRC1 and TGFβ1 treatment (1 h) on SMAD3 phosphorylation in beige hMADS cells in presence of IgG and neutralizing anti-TGFβ antibody ($n = 6$; $p = 0.0133$ ctrl vs CTHRC1 within IgG, $p < 0.0001$ ctrl vs TGFβ1 within IgG, $p < 0.0001$ CTHRC1 vs TGFβ1 within IgG, $p = 0.0442$ ctrl vs CTHRC1 within anti-TGFβ, $p < 0.0001$ ctrl vs TGFβ1 within anti-TGFβ and $p = 0.0095$ CTHRC1 vs TGFβ1 within anti-TGFβ). **g** Representative images with scale bar 100 μm and quantification of **h** phosphorylated SMAD3 (green) positive nuclei (blue) ($n = 15$; $p = 0.0002$ for CTHRC1 and $p < 0.0001$ for TGFβ1) and **i** total SMAD3 (green) shuttling in response to CTHRC1 and TGFβ1 ($n = 1400$; $p < 0.0001$ for both CTHRC1 and TGFβ1). Experiment was repeated independently twice with similar results. **j** Representative western blots and quantification of the binding of CTHRC1 tagged with HiBiT in control and stable *GPR180* knockout HEK-293T cells ($n = 6$; $p < 0.0001$). **k** Effect of CTHRC1 stimulus (1 h) on SMAD3 phosphorylation in TGFβR1 kinase inhibitor pre-treated cells ($n = 4$; $p = 0.0356$ for CTHRC1, $p < 0.0001$ for TGFβ1 and $p < 0.0001$ for CTHRC1 vs TGFβ1 within DMSO). SMAD3 phosphorylation in response to CTHRC1 in combination with silencing of **l** *TGFβR1* ($n = 6$; $p = 0.0053$ for PBS ctrl siRNA vs CTHRC1 ctrl siRNA, $p = 0.0137$ for CTHRC1 ctrl siRNA vs CTHRC1 si*TGFβR1*) and **m** *TGFβR2* ($n = 8$; $p < 0.0001$ for PBS ctrl siRNA vs CTHRC1 ctrl siRNA, $p < 0.0001$ for CTHRC1 ctrl siRNA vs CTHRC1 si*TGFβR2* and $p = 0.0275$ for PBS si*TGFβR2* vs CTHRC1 si*TGFβR2*) in human beige adipocytes. **n** Luciferase activity in naïve and *GPR180* knockout HEK-293T cells transfected with plasmid expressing 4× SMAD binding elements (SBE) upstream of luciferase in response to CTHRC1 and TGFβ1 stimulus for 18 h ($n = 4$; $p < 0.0001$ for both treatments in naïve HEK-293T cells and $p = 0.0261$ for TGFβ1 in GPR180 ablated HEK293Ts cells). **o** SMAD3 phosphorylation in beige hMADS cells treated with different concentrations of TGFβ1 in combination with CTHRC1 ($n = 3$; $p < 0.0001$ for 300 ng/ml, $p = 0.0362$ for 100 ng/ml and $p = 0.0177$ for 3000 ng/ml). **p** SMAD3 phosphorylation in response to TGFβ1 (300 pg/ml) in combination with increasing dose of CTHRC1 ($n = 3$; $p = 0.0003$ for 100 ng/ml, $p = 0.0099$ for 300 ng/ml, $p < 0.0001$ for 1000 ng/ml and $p = 0.0004$ for 3000 ng/ml). **q** Phosphorylation of SMAD3 in response to CTHRC1 and TGFβ1 ($n = 3$; CTHRC1 $p = 0.0021$, TGFβ 1 pg/ml $p = 0.0281$ and 10 pg/ml $p < 0.0001$). **r** Regulation of CTHRC1 in beige adipocytes following TGFβ1 treatment ($n = 6$; $p < 0.0001$ for both 0.1 and 1 ng/ml). Data are shown as average ±SEM. Statistical analysis was performed by two-sided Student's *t*-test (**j**, **o**, **p**), one-way (**e**, **h**, **i**, **q**, **r**) or two-way ANOVA (**c**, **d**, **f**, **k–m**) with Dunnett's or Tukey post-hoc test, respectively. Significant difference is indicated as *$p < 0.05$, **$p < 0.01$ and ***$p < 0.001$. CTHRC1 Collagen triple helix repeat containing 1, DMSO dimethyl sulfoxide, GPR180 G protein-coupled receptor 180, hMADS human multipotent adipose-derived stem cells, IgG immunoglobulin G, n.d. not detected, PBS phosphate buffered saline, RFP red fluorescent protein, SMAD3 Mothers against decapentaplegic homolog 3, TGFβ1 Transforming growth factor β1, TGFβR1 Transforming growth factor β receptor type 1, TGFβR2 Transforming growth factor β receptor type 2.

an increase in cumulative energy expenditure and a shift in RER towards fatty acid oxidation (Fig. 6i and j). Although we utilized a liver-specific promoter to overexpress *Cthrc1*, we did not notice any detrimental effects of the overexpression, since liver mass (Supplementary Fig. 6f), plasma ALT activity (Supplementary Fig. 6g) or expression of inflammatory (Supplementary Fig. 6h) and fibrotic (Supplementary Fig. 6i) markers were either unchanged or decreased. Importantly, we observed an improved glucose tolerance in mice with elevated CTHRC1 (Fig. 6k), which was independent of body weight loss (Supplementary Fig. 6j and k) and was observed only in wild-type but not in global *Gpr180* knockout mice, further supporting our in vitro findings that GPR180 mediates the beneficial metabolic effects of CTHRC1.

**Human GPR180/CTHRC1 is associated with an improved metabolic profile**. To study the relevance of these two components of TGFβ signalling pathway in humans, we analysed the regulation of GPR180 and CTHRC1 in participants with normal weight, with obesity and normal glucose tolerance (NGT), with obesity and prediabetes and in individuals with obesity and newly diagnosed type 2 diabetes. The expression of *GPR180* in sub-cutaneous WAT was significantly lower in participants with obesity, independent of glycaemic control (Fig. 7a). In a subset of study participants, for which we had paired samples of adipose tissue and isolated mature adipocyte fraction, we could show that the expression of *GPR180* was 2.5-fold higher in adipocytes than in the whole tissue (Fig. 7b), indicating that adipocytes account to large extent for *GPR180* tissue level. Interestingly, we observed a negative correlation of *GPR180* expression in WAT with fat cell size (Fig. 7c) and suppression of fatty acids release during euglycemic hyperinsulinemic clamp, reflecting the adipocyte insulin sensitivity (Fig. 7d). Importantly, adipose tissue *GPR180* expression positively correlated with basal resting energy

expenditure (REE) (Fig. 7e). Next, we measured plasma concentration of CTHRC1 in the same cohort of men. Since the circulating levels of this hormone are low, in several individuals the levels were below the detection limit (sensitivity of the assay 30 pg/ml; lower limit of detection 10 pg/ml). When we stratified the whole measured population into two subgroups based on the CTHRC1 levels, we found that individuals with higher CTHRC1 are younger and more metabolically healthy as evidenced by a lower BMI, smaller adipocyte size, higher insulin sensitivity and better plasma lipid profile as well as lower ectopic lipid deposition (Supplementary Tab. 1). In line with this observation, CTHRC1 prevalence was significantly lower in individuals with diabetes (Fig. 7f). In addition, a positive association of circulating CTHRC1 and energy expenditure was observed (Fig. 7g and h). In conclusion, our data indicate that GPR180 and CTHRC1 are metabolically relevant also in humans and might play an important role in regulation of adipose tissue and whole-body energy metabolism.

Taken together, we show here that CTHRC1 requires GPR180 to activate TGFβ signalling pathway (Fig. 7i), which improves metabolic control by promoting adipocyte thermogenic activity and increasing energy expenditure.

## Discussion
BAT is considered an important regulator of systemic glucose and lipid homeostasis[6,10], therefore, its activation represents a prospective strategy to increase energy expenditure and ameliorate metabolic diseases. In an unbiased transcriptomic analysis of human brown and white adipose tissue samples, we identified GPR180 as a receptor enriched in human BAT compared to WAT and demonstrated its significance in the regulation of brown and beige adipocyte function and glucose homeostasis by utilizing genetic and pharmacological tools. Mechanistically, we

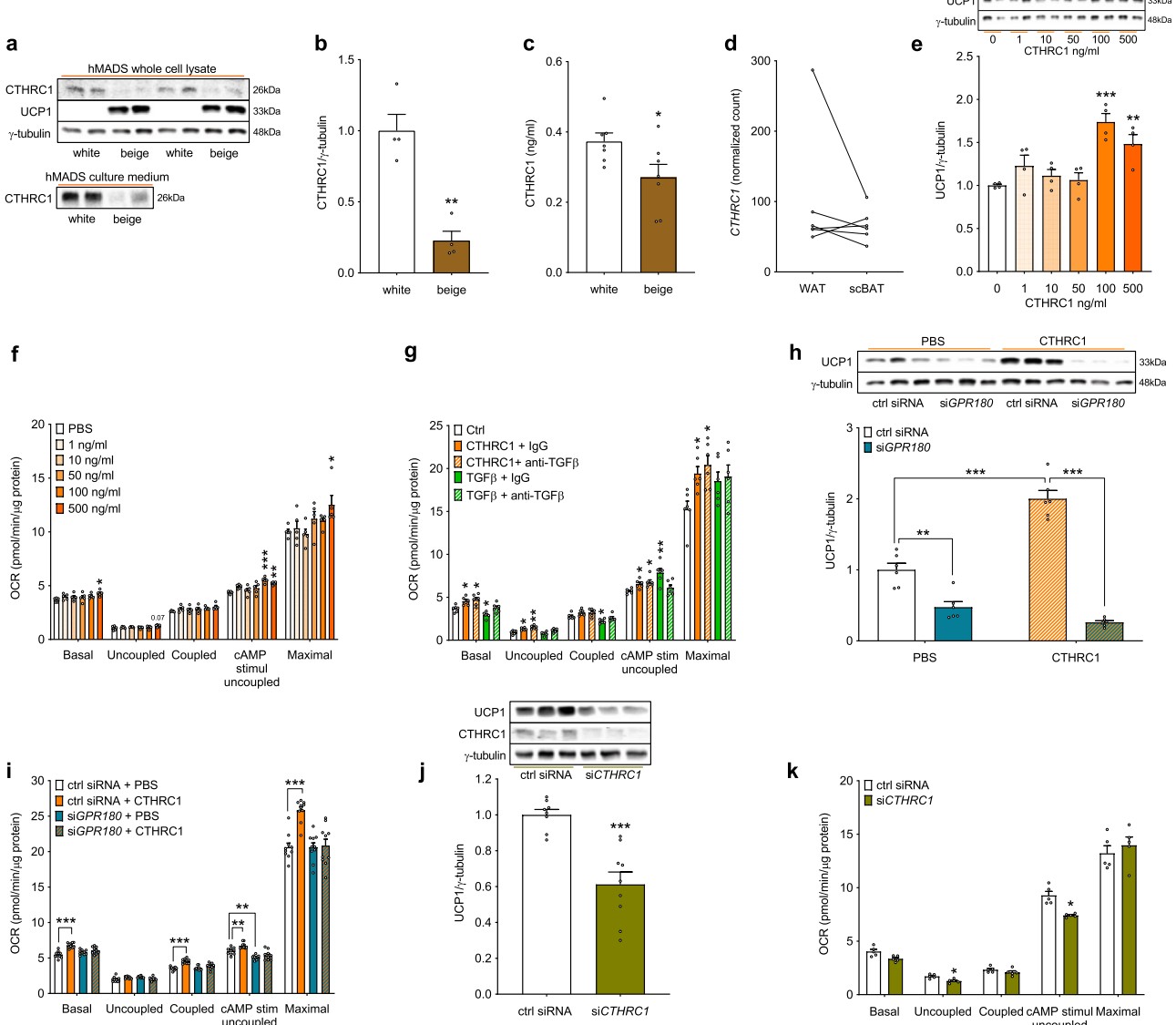

**Fig. 5 CTHRC1, an adipokine that requires GPR180 to enhance the beige adipocyte phenotype. a** Representative blots and **b** quantification of CTHRC1 protein in human adipocytes ($n = 4$; $p = 0.0011$) and **c** cell-conditioned media ($n = 7$; $p = 0.0408$). **d** Expression of *CTHRC1* in supraclavicular brown and subcutaneous white adipose tissue of 6 healthy volunteers with detectable BAT ($p = 0.3008$). Effect of long-term (72 h) CTHRC1 treatment on **e** UCP1 protein ($n = 4$; $p = 0.0001$ for 100 ng/ml and $p = 0.0064$ for 500 ng/ml) and **f** mitochondrial respiration ($n = 5$; basal $p = 0.0102$, cAMP-stimulated uncoupled $p < 0.0001$ for 100 ng/ml and $p = 0.0021$ for 500 ng/ml, maximal $p = 0.0207$) in mature human beige adipocytes. **g** Effect of long-term (72 h) CTHRC1 (500 ng/ml) and TGFβ1 (1 ng/ml) on mitochondrial respiration in beige hMADS cells in presence of IgG (control) and neutralizing anti-TGFβ antibody (1 μg/ml) ($n = 6$; basal CTHRC1 IgG $p = 0.0376$, CTHRC1 anti-TGFβ $p = 0.0489$, TGFβ IgG $p = 0.0474$; uncoupled CTHRC1 IgG $p = 0.0372$, CTHRC1 anti-TGFβ $p = 0.0079$; coupled TGFβ IgG $p = 0.0449$; cAMP-stimulated uncoupled CTHRC1 IgG $p = 0.0407$, CTHRC1 anti-TGFβ $p = 0.0440$, TGFβ IgG $p = 0.0317$; maximal CTHRC1 IgG $p = 0.0478$, CTHRC1 anti-TGFβ $p = 0.0310$). **h** UCP1 levels ($n = 6$; si*GPR180* in PBS $p = 0.0016$, CTHRC1 in ctrl siRNA $p < 0.0001$ and si*GPR180* in CTHRC1 treatment $p < 0.0001$) and **i** mitochondrial respiration ($n = 10$; $p < 0.0001$ basal, $p < 0.0001$ coupled, $p = 0.0094$ for CTHRC1 treatment in ctrl siRNA and $p = 0.0013$ for si*GPR180* in PBS treated in cAMP-stimulated uncoupled respiration, $p < 0.0001$ maximal respiration) in adipocytes following long-term (72 h) CTHRC1 treatment (500 ng/ml) in combination with *GPR180* knockdown. Effect of *CTHRC1* silencing on (j) UCP1 protein (ctrl siRNA $n = 8$, si*CTHRC1* $n = 9$; $p = 0.0002$) and **k** mitochondrial oxygen consumption ($n = 5$; $p = 0.0311$ for uncoupled respiration, $p = 0.0326$ for cAMP-stimulated uncoupled respiration) in hMADS cells. Data are shown as mean ± SEM. Statistical analysis was performed by unpaired Student´s *t*-test (**b**, **c**, **j**), paired Student´s *t*-test (**d**), one-way ANOVA with Dunett's post-hoc-test (**e**) or two-way ANOVA with Sidak and Tukey post-hoc tests (**f–i**, **k**). Significance is indicated as \**p* < 0.05, \*\**p* < 0.01 and \*\*\**p* < 0.001. cAMP cyclic adenosine monophosphate, CTHRC1 Collagen triple helix repeat containg 1, GPR180 G protein-coupled receptor 180, hMADS human multipotent adipose-derived stem cells, IgG imunoglobulin G, OCR oxygen consumption rate, PBS phosphate buffered saline, scBAT supraclavicular brown adipose tissue, TGFβ Transforming growth factor β, UCP1 Uncoupling protein 1, WAT white adipose tissue.

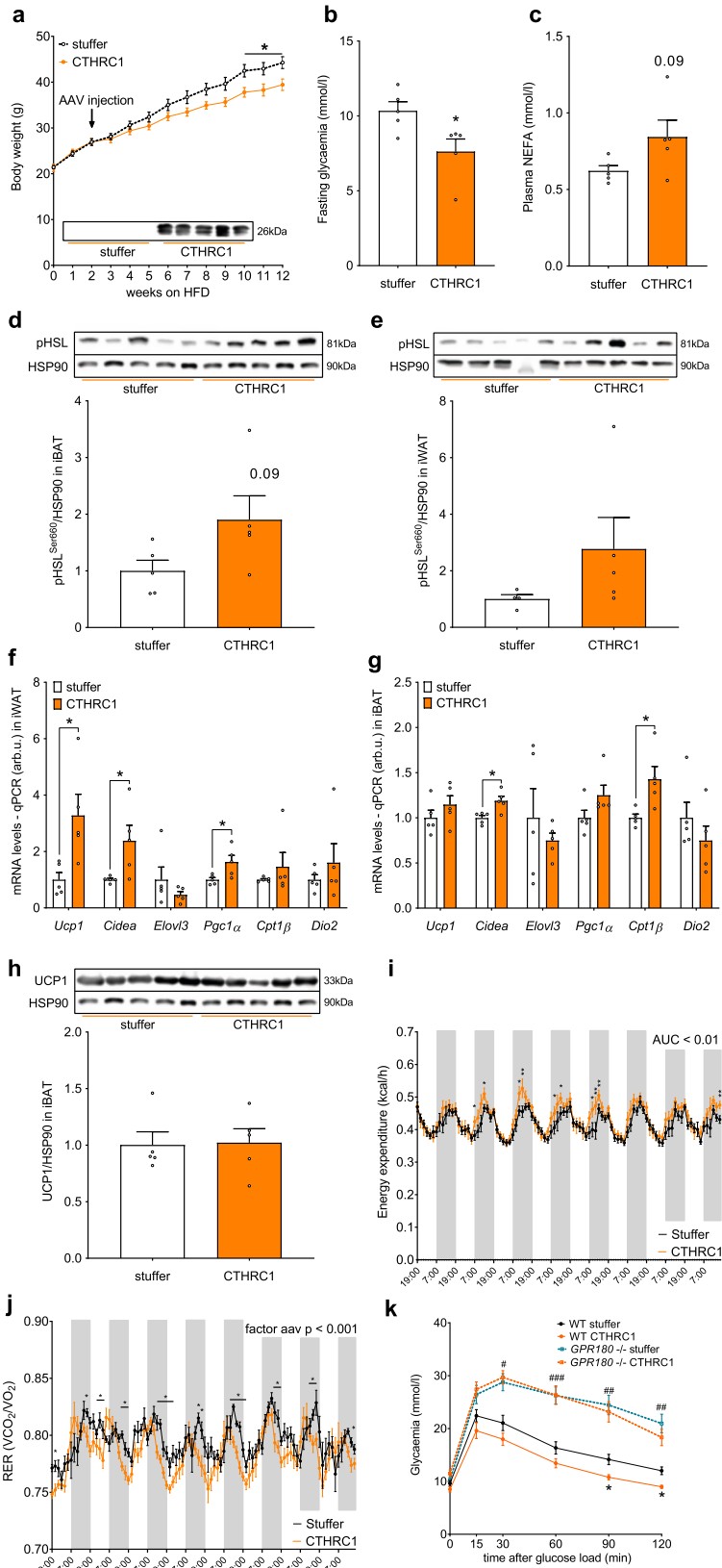

show that GPR180 is in fact not a classical GPCR, but a component of TGFβ signalling hub, which binds and mediates the effects of the circulating factor CTHRC1 to regulate thermogenic adipocyte function. Furthermore, we provide evidence that both components are physiologically relevant and participate on the regulation of the metabolic phenotype in mouse and human.

GPCRs are structurally characterized by seven transmembrane domains with extracellular N-terminus and intracellular C-terminus localization[26,27]. GPR180 was originally described as a Rhodopsin-like orphan GPCR involved in vascular remodelling[28]. Interestingly, several highly conserved motifs characteristic for GPCRs such as tripeptide (D/E)R(Y/W) at the intracellular end of transmembrane

**Fig. 6 CTHRC1 requires GPR180 to ameliorate metabolic disturbances in obesity.** Effect of AAV-mediated CTHRC1 overexpression on **a** body weight ($p = 0.0174$ week 10, $p = 0.0191$ week 11 and $p = 0.0133$ week 12), **b** fasting blood glucose ($p = 0.03070$ and **c** circulating FFA in 20-weeks-old male C57Bl/6 N mice challenged with HFD ($n = 5$). HSL phosphorylation (at Serine 660) in **d** iBAT and **e** iWAT of mice with increased circulating CTHRC1 ($n = 5$). Gene expression of selected brown adipocyte markers in **f** iWAT ($p = 0.0280$ for *Ucp1*, $p = 0.0377$ for *Cidea* and $p = 0.0320$ for *Pgc1α*) and **g** iBAT ($p = 0.0111$ for *Cidea* and $p = 0.0191$ for *Cpt1β*) of animals overexpressing CTHRC1 ($n = 5$). **h** UCP1 protein in iBAT following CTHRC1 treatment in male mice exposed to HFD. Effect of CTHRC1 overexpression on **i** energy expenditure (AUC $p = 0.0014$) and **j** respiratory exchange ratio (effect of CTHRC1 overexpression $p = 0.0005$) in C57Bl/6 N mice challenged with HFD AAV was injected prior to acclimatization in metabolic cages and measurement ($n = 7$). **k** Intraperitoneal glucose tolerance test in wild-type and *GPR180* knockout mice fed with HFD and overexpressing stuffer or CTHRC1 ($n = 8$-$9$; $p = 0.0160$ for *Gpr180*$^{-/-}$ vs WT at 30 min, $p = 0.0011$ for *Gpr180*$^{-/-}$ vs WT at 60 min, $p = 0.0020$ for *Gpr180*$^{-/-}$ vs WT at 90 min, $p = 0.0366$ for WT stuffer vs WT CTHRC1 at 90 min, $p = 0.0041$ for *Gpr180*$^{-/-}$ vs WT at 120 min, $p = 0.0160$ for WT stuffer vs WT CTHRC1 at 120 min). Data are presented as mean ± SEM. Statistical significance was calculated using two-sided Student´s *t*-test (**b**–**j**) or two-way ANOVA with repeated measurements followed by Sidak post-hoc test (**a**, **j**, **k**). Area under the curve was calculated to compare energy expenditure in CTHRC1 overexpressing mice (**i**). Statistical differences are indicated as *$p < 0.05$; (**k**) * WT CTHRC1 vs WT stuffer, # WT stuffer vs GPR180$^{-/-}$ stuffer, $^{\#}p < 0.05$, $^{\#\#}p < 0.01$ and $^{\#\#\#}p < 0.001$. AAV adeno-associated virus, AUC area under the curve, CIDEA Cell death inducing DFFA like effector a, CPT1β Carnitine palmitoyltransferase 1β, CTHRC1 Collagen triple helix repeat containing 1, DIO2 Iodothyronine deiodinase 2, ELOVL3 Fatty acid elongase 3, GPR180 G protein-coupled receptor 180, HFD high-fat diet, HSL Hormone sensitive lipase, HSP90 Heat shock protein 90, iBAT interscapular brown adipose tissue, iWAT inguinal white adipose tissue, NEFA non-esterified fatty acids, PGC1A PPARγ coactivator 1α, RER respiratory exchange ratio, UCP1 Uncoupling protein 1, WT wild-type.

helix III or NPXXY sequence near cytoplasmic end in helix VII[27] are absent in the GPR180 sequence. Besides the fact that we did not observe any changes in G-protein signalling, we provide evidence that GPR180 has a reverse orientation in the plasma membrane which is similar to the adiponectin receptor[29]. This is further supported by the lack of GXXXN motif in the first transmembrane domain of GPR180 that was recently discovered to be present in many GPCRs and being one of the mechanisms responsible for canonical orientation in the plasma membrane[30].

The TGFβ signalling pathway is involved in regulation of diverse physiological processes such as cellular proliferation, differentiation and growth, immune response, extracellular matrix deposition and tissue repair[31]. TGFβ pleiotropy is mainly based on its immunosuppressive and pro-fibrogenic properties. Deficiency in TGFβR1 and TGFβR2 results into progressive wasting syndrome or embryonal lethality, respectively[32,33]. Conditional knockout of both receptors in different tissues or global deletion of SMAD3 is associated with increased tumorigenesis[14,15,34] demonstrating the importance of this signalling pathway in cellular homeostasis. Defective TGFβ signalling due to increased expression of inhibitory SMAD7 leads to attenuation of SMAD3 phosphorylation, which is characteristic for gut mucosa of patients with Crohn´s disease[35]. Moreover, antisense oligonucleotide based therapy targeting SMAD7 seems to be effective in restoring TGFβ signalling and suppressing mucosal inflammation in these patients[36]. Interestingly, SMAD3 deficient mice are protected from obesity and diabetes[37], however, these mice display striking differences in body weight and length already at 8 weeks of age indicating developmental defects. On the other hand, it is well established that TGFβ pathway over-activation results in excessive accumulation of extracellular matrix and tissue fibrosis, which is associated with organ dysfunction[38]. Moreover, tumour progression is associated with transformation of cellular response to TGFβ and blockade of the pathway is beneficial[39]. Altogether, these data indicate that intermediate response or fine-tuning of the TGFβ signalling pathway, in agreement with the concept of hormesis[40], is essential to maintain whole-body homeostasis.

The TGFβ signalling machinery is complex with multiple levels of regulation to achieve signal transduction. The simple model of TGFβR1 and TGFβR2 dimerization in response to ligand is outdated since multiple co-receptors regulating ligand presentation, availability and ligand-independent pathway activation have been identified in the last years[19]. Such an example is GPR50, a GPCR that forms an alternative complex with TGFβR1 in a TGFβR2-independent manner to enhance its basal capacity to induce SMAD3 phosphorylation via interaction with FKBP12[41]. Identification of GPR180 and CTHRC1 as components of the TGFβ signalling pathway activating SMAD3 further increases the complexity of the TGFβ signalling pathway and provides the first adaptive pathway that is involved in low-grade activation of the signalling pathway and modulation of TGFβ response. In addition, we show here that GPR180 is required for full activation of the TGFβ signalling machinery; however, even submaximal activation is sufficient to drive browning of the cells. Currently, we do not know whether GPR180 is a downstream modulator of TGFβ activity, or whether it directly interacts with either the whole receptor complex or solely with the TGFβR1 kinase. Given that the induction of SMAD3 signalling by TGFβ1 is blunted in presence of CTHRC1, together with our finding that the effect of CTHRC1 is dependent at least partially on both TGFβ receptors, argue in favour of a function as a components of TGFβ receptor machinery.

Several studies have linked CTHRC1 to TGFβ signalling albeit only in an intracellular context[42,43]. According to these reports, CTHRC1 directly interacts with and accelerates the degradation of phospho-SMAD3 in the cytosol of rat PAC1 smooth muscle cells and human stellate LX-2 cells, indicating a cytosolic action of endogenous CTHRC1. This is in stark contrast to the function of extracellular or circulating CTHRC1, which targets GPR180 and TGFβR1. Here we demonstrate that extracellular recombinant CTHRC1 increases phosphorylation of SMAD3 as well as its shuttling into the nucleus in beige hMADS-derived adipocytes and HEK-293T cells by activating GPR180. Similar observations were made following treatment of hepatic stellate cells (HSCs)[44]. Interestingly, CTHRC1 expression is regulated by TGFβ in fibroblasts and HSCs since the SMAD binding elements were identified in the promoter of *Cthrc1* gene[42,45], and here we confirm the responsiveness of CTHRC1 to TGFβ1 in mature human adipocytes. These data suggest the existence of a feedback loop, in which CTHRC1 upregulated in response to TGFβ activation tightly controls intracellular stability of SMAD3 as well as fine-tunes high TGFβ stimulatory potential, thereby maintaining only a certain level of SMAD3 phosphorylation. In addition, it should be noted that CTHRC1 induces only a submaximal SMAD3 phosphorylation, suggesting that CTHRC1 and GPR180 represent an alternative axis to achieve a low-grade activation of TGFβ signalling within a hormetic range, which is required to maintain physiological functionality of the tissue and prevent its pathological remodelling.

As mentioned above, the TGFβ pathway is involved in regulation of cellular proliferation and differentiation. Recently, the

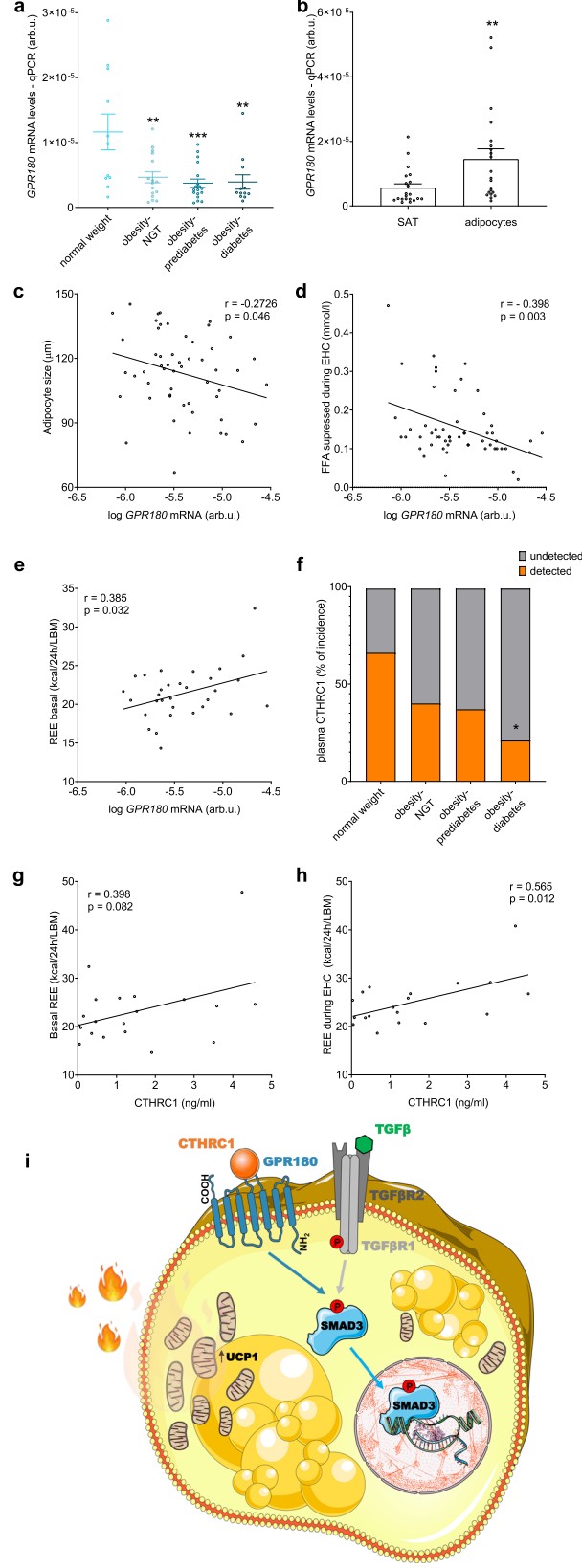

**Fig. 7 Regulation of GPR180 and CTHRC1 in human. a** Expression of *GPR180* in WAT of normal weight men and individuals with obesity and normal glucose tolerance (NGT), obesity with prediabetes and obesity with diabetes (n = 57; p = 0.0021 for individuals with obesity and NGT, p = 0.0003 for individuals with obesity and prediabetes and p = 0.0013 for patients with obesity and diabetes). **b** Expression of *GPR180* in paired samples of SAT and isolated adipocytes (n = 21; p = 0.0077). Correlation of adipocyte *GPR180* level with **c** adipocyte size (n = 54), **d** suppression of fatty acid release during EHC (n = 55) and **e** resting energy expenditure (n = 31). **f** Incidence of circulating CTHRC1 in normal weight individuals, and patients affected by obesitybut NGT, patients having obesity and prediabetes and/or type 2 diabetes (n = 85; p = 0.02533) and its association with **g** basal energy expenditure and **h** energy expenditure measured during the steady state of EHC (n = 20). **i** Schematic illustration of the signalling mechanism identified in this study. Data are shown as mean ± SEM. Statistical analysis was performed by one-way ANOVA with Dunett post-hoc test (**a**), paired Student´s t-test (**b**) or Fisher´s exact test (**f**). For association of *GPR180* expression in WAT or circulating CTHRC1 with metabolic parameters Pearson's correlation coefficient was calculated (**c**–**e**, **g**, **h**). Significant differences are indicated as *p < 0.05, **p < 0.01 and ***p < 0.001. CTHRC1 Collagen triple helix repeat containing 1, EHC euglycaemic hyperinsulinaemic clamp, FFA free fatty acids, GPR180 G protein-coupled receptor 180, REE resting energy expenditure, SAT subcutaneous adipose tissue, SMAD3 Mothers against decapentaplegic homolog 3, TGFβ1 Transforming growth factor β1, TGFβR1 Transforming growth factor β receptor type 1, TGFβR2 Transforming growth factor β receptor type 2, UCP1 Uncoupling protein 1.

CTHRC1 acting through GPR180, does not affect formation of human adipocytes, but specifically promotes brown adipocyte activity and adipocyte browning, which could be attributed to low GPR180 expression in progenitor cells and its abundance in mature adipocytes. Activin, another ligand of TGFβ family which can signal via SMAD3, was shown to enhance UCP1 expression in mature murine adipocytes[46] and to increase energy expenditure by stimulation of BAT thermogenesis[47]. Interestingly, TGFβ2 treatment was recently also shown to upregulate UCP1 expression and increase glucose uptake in BAT[48]. These opposing effects of TGFβ signalling on progenitors and mature adipocytes are not unusual. For example Janus kinase[16] or Rho kinase[49,50] display similar behaviours.

Attenuated CTHRC1 signalling as a consequence of GPR180 ablation was associated with deterioration of the metabolic profile, while CTHRC1 administration exerts protective effects. This is in line with the phenotype of CTHRC1 null mice, which display increased fat mass, liver steatosis and reduced energy expenditure[51,52]. Although we cannot exclude the contribution of other organs to the observed phenotype due to systemic elevation of CTHRC1 and ubiquitous expression of GPR180, we clearly show that adipose tissue thermogenic activity is the key effector of this phenotype, which is supported by the adipocyte-specific knockout and in vitro functional studies. In addition, our data indicate that the observed changes in adipose tissue thermogenic activity cannot be attributed exclusively to alterations in UCP1 content, but might rather involve changes in alternative thermogenic mechanisms, such as fatty acid flux. Therefore, targeting GPR180 could improve glucose tolerance and modulate body weight by promoting adipose tissue thermogenic activity, although CTHRC1 mediated low-grade SMAD3 activation is clearly not the main driver of this process. Even modest changes in energy expenditure in response to CTHRC1/GPR180 manipulation have a profound effect on whole-body energy metabolism in a long term, similarly to other interventions[53,54]. However, the potential of CTHRC1/GPR180 axis to affect metabolic health is

role of this pathway in adipogenic commitment and thermogenic adipocyte differentiation was extensively investigated and these studies unanimously demonstrated inhibitory effect of activated TGFβ signalling on recruitment and activation of beige/brite adipocyte progenitors[16–18]. However, our data indicate that

further supported by the positive association of both ligand and receptor with energy expenditure in humans. We and others[55] have shown that circulating levels of CTHRC1 in human population are low, supporting para- or autocrine action of this secreted factor. While the percentage of CTHRC1 detectability in plasma is similar in our and Duarte study, there is a discrepancy in its regulation in type 2 diabetes. This could be due to differences in sample processing time, which was within 30 m in our cohort in contrast to 6–24 h in the other work[55]. Moreover, it is important to note that type 2 diabetic individuals recruited in our study were newly diagnosed and did not receive any medication at the time of examination.

In conclusion, we identify here GPR180 and CTHRC1 as components of TGFβ signalling pathway, which regulate low-grade SMAD3 phosphorylation required for proper thermogenic adipocyte function and control of whole-body energy and glucose homeostasis.

## Methods

**Contact for reagent and resource sharing**. Further information and requests for resources and reagents should be directed to and will be fulfilled by the Lead Contact, Professor Christian Wolfrum (christian-wolfrum@ethz.ch).

**Clinical transcriptome study**. The clinical study for cohort 1 was approved by the Ethics Committee of the Hospital District of Southwest Finland and conducted according to the principles of the Declaration of Helsinki. All study participants provided written consent prior to entering the study. The subjects were screened for medical history and status, and only healthy volunteers were enrolled in the study. PET-CT scan and tissue biopsies were performed as described in the previous publication[56]. Briefly, the subjects underwent a PET-CT examination after an overnight fast. On the cold exposure day, the subjects spent 2 h wearing light clothing in a room with an ambient temperature of $17 \pm 1\,°C$ before moving into the PET-CT room, which had an air temperature of 23 °C. During the PET-CT session, one foot of the subject was placed intermittently (5 min in/5 min out) in cold water at a temperature of $8 \pm 1\,°C$. Detailed description of PET-CT examination is available in[56]. The site of the biopsy was selected based on the cold exposure [18]FDG-PET-CT image that showed activated BAT. A subcutaneous WAT sample was collected from the same incision. The biopsies were obtained under local lidocaine anaesthesia by a plastic surgeon at normal room temperature (20 °C) one week after the PET-CT examination. Immediately after removal, the tissue samples were snap frozen in liquid nitrogen and stored at −80 °C until further processing. The RNA from adipose tissue was isolated using the RNeasy Lipid Tissue Mini Kit (QIAGEN), according to the manufacturer's protocol including the DNase treatment step. Primary outcomes of this study have been previously published.

**Study with normal weight participants and participants with obesity and differing glycaemic control**. The clinical study was approved by the Local Ethics Committee of the University Hospital in Bratislava, Slovakia and it conforms to the ethical guidelines of the 2000 Helsinki declaration. All study participants provided witnessed written informed consent prior entering the study. Eighty-five middle-aged sedentary men were recruited. Screening measurements of BMI, fasting, and/or 2-h glycemia (oGTT) were applied to stratify them into following groups: (i) metabolically healthy normal weight individuals and patients with similar levels of obesity and (ii) normal glucose tolerance; (iii) prediabetes and (iiii) newly diagnosed, yet untreated, type 2 diabetes. Patients with chronic disease or regular use of pharmacotherapy were excluded. Subcutaneous adipose tissue samples were taken by aspiration with needle biopsy from abdominal region in the fasted state under local subcutaneous anesthesia (1% Mesokain, Leciva, Pragues, Czech Republic). The sample was quickly washed in saline to eliminate blood and frozen in liquid nitrogen prior RNA extraction. Patients underwent complex metabolic phenotyping[57].

**Mouse experiments**. All authors of the study who participated in mouse experiments have complied with ethical regulations for animal testing and research. All animal procedures in this study were approved by cantonal ethics committee of the veterinary office of the Canton of Zürich. Sample size was determined based on previous experiments in our lab and similar studies reported in the literature. All mice used for the experiments were male, housed 3–4 littermates per cage in individually ventilated cages at standard housing conditions (22 °C, 12 h reversed light/dark cycle, dark phase starting at 7am, 40% humidity), with *ad libitum* access to chow (18% proteins, 4.5% fibers, 4.5% fat, 6.3% ashes, Provimi Kliba SA) and water. Health status of all mouse lines were regularly monitored according to FELASA guidelines.

**Generation of the GPR180 global knockout**. The *GPR180* null allele was obtained by Cas9/CRISPR. The Cas9/CRISPR target sequence tgactcagagagcccccagg(ggg) (PAM sequence in brackets) in exon3 of the *GPR180* gene was modified directly in mouse one-cell embryos by electroporation. C57BL/6 J female mice underwent ovulation induction by i.p. injection of 5 IU equine chorionic gonadotrophin (PMSG; Folligon–InterVet), followed by i.p. injection of 5 IU human chorionic gonadotropin (Pregnyl–Essex Chemie) 48 h later. For the recovery of embryos, C57BL/6 J females were mated with males of the same strain immediately after the administration of human chorionic gonadotropin. Embryos were collected from oviducts 24 h after the human chorionic gonadotropin injection and were then freed from any remaining cumulus cells by a 1–2 min treatment of 0.1% hyaluronidase (Sigma–Aldrich) dissolved in M2 medium (Sigma). Prior to electroporation, the *zona pellucida* was partially removed by brief treatment with acid Tyrode's solution and the embryos were washed and briefly cultured in M16 (Sigma) medium at 37 °C and 5% $CO_2$. Electroporation with a mixture of 16uM cr:trcrRNA hybrid targeting *GPR180* and 16uM Cas9 protein (all reagents from IDT) was carried out using 1 mm gap electroporation cuvette and the ECM830 electroporator (BTX Harvard Apparatus). Two square 3 ms pulses of 30 V with 100 ms interval were applied as these conditions provide efficient genome editing with embryonic survival[58]. Surviving embryos were washed with M16 medium and transferred immediately into the oviducts of 8–16-wk-old pseudopregnant Crl:CD1(ICR) females that had been mated with sterile genetically vasectomized males[59] the day before embryo transfer (0.5 dpc). Pregnant females were allowed to deliver and raise the pups until weaning age. In total 100 embryos were electroporated and 97 surviving embryos were transferred into 4 foster mothers. All foster mothers produced live litters with a total of 26 viable F0 pups. Five F0 pups carried in/del modification in the *GPR180* gene.

**Generation of the adipocyte-specific inducible GPR180 knockout**. The *GPR180*[fl/fl] mouse strain used for this research project was created from embryonic stem (ES) cell clone EPD0539_4_D02, obtained from the KOMP Repository (www.komp.org) and generated by the Wellcome Trust Sanger Institute (WTSI). Targeting vectors used were generated by the WTSI and the Children's Hospital Oakland Research Institute as part of the Knockout Mouse Project (3U01HG004080)[60]. Chimeric mice were generated by injection of ES into blastocyst in ETH Phenomics Center Zurich (epic.ethz.ch). Conditional allele was achieved via flippase (Flp) recombination of knockout first allele. Flp deleter mouse was kindly provided by Prof. Markus Stoffel (ETH Zurich). Inducible adipocyte-specific ablation of GPR180 was completed after crossing *GPR180*[fl/fl] mice to *Adip*-CreERT2 mice[4]. At 12 weeks of age, recombination of the floxed allele was induced by oral tamoxifen gavage (2 mg/mouse in sunflower oil, Sigma–Aldrich).

**Adeno-associated virus (AAV) to overexpress CTHRC1**. The expression cassette consisting of a Kozak sequence followed by the coding sequence for murine *CTHRC1* (UniProt Q9D1D6) was codon-usage optimized for expression in mice and synthesized (Thermo Fisher Scientific/Geneart, Regensburg, Germany) before being cloned into a pFB vector. The vector contains AAV2 ITRs, from which one is lacking the terminal resolution site and a LP-1 promoter[61] to drive liver-specific protein expression. As a control a pFB-Stuffer vector[62] was used for rAAV production. Recombinant AAV8 vectors were produced by calcium phosphate transfection of human embryonic kidney HEK-293T cells using the pFB_LP1-mCTHRC1 or the pFB_Stuffer plasmid in combination with the pDP8 (Plasmid Factory, Bielefeld, Germany) and the pHelper plasmid (Thermo Fisher Scientific), followed by purification based on polyethyleneglycol precipitation (PEG-8000 solution), iodixanol gradient and ultrafiltration (Amicon Ultra-15 MWCO 100,000 ultrafiltration tube; Merck Millipore). Titer was determined by quantitative PCR. Male C57BL/6 N mice (Charles River) and/or *GPR180* global knockouts and their wild-type littermates were injected i.v. with CTHRC1 AAV or stuffer (non-coding DNA), both at 5.00E + 10 VG/mouse.

In high-fat diet cohorts (23.9% proteins, 4.9% fibers, 35% fat, 5.0% ashes, Provimi Kliba SA), the feeding regimen (12 weeks) was initiated at age of 8–10 weeks. Thermoneutrality cohort was housed at 30 °C after mice weaning at age of 3–4 weeks to suppress brown adipose tissue activity and subcutaneous white fat browning prior glucose tolerance test at age 12 weeks.

**Cell culture—hMADS cells**. HMADS cells originating from the prepubic fat pad of a 4-month-old male were kindly provided by Dr. Amri (Elabd et al., 2009). Cells (between passage 14 and 16) were grown in low glucose DMEM supplemented with 15 mM HEPES, 10% FBS, 2mM L-glutamine, 1% Penicillin/Streptomycin and 2.5 ng/ml recombinant human FGF-2 (Peprotech) in normoxic humidified cell culture incubator (5% $CO_2$ and 37 °C). The medium was changed every other day and FGF-2 was omitted after cells reached confluence. Differentiation of 48 h post-confluent cells was induced (day 0) by adipogenic medium (DMEM/Ham's F12 media (Lonza) containing 10 μg/ml Transferrin, 10 nM insulin and 0.2 nM triiodothyronine) supplemented with 1 μM dexamethasone and 500 μM isobutyl methylxanthine (IBMX) and from day 2 to 9, cells were cultured in adipogenic medium containing 100 nM rosiglitazone. All compounds were obtained from Sigma–Aldrich except for rosiglitazone (Adipogen). Cells were kept in culture until day 18 in absence of rosiglitazone to obtain mature white adipocytes. To obtain

beige adipocytes, cells were exposed to an additional rosiglitazone pulse (100 nM) between days 14 and 18. To investigate long-term effect of pharmacological agents on browning of white adipocytes, treatment was performed for three consecutive days in combination with rosiglitazone. Signalling studies were performed on day 18 in mature beige hMADS cells-derived adipocytes after 2 h fasting in adipogenic medium without supplements following acute stimulus with recombinant proteins up to 1 h. Recombinant CTHRC1 was produced by human cells and purchased from 2 different vendors (Sino Biological and Creative Biomart). Treatment with both recombinant proteins yielded the same results. Recombinant FSTL1 (Abcam) was produced by CHO cells and IGFBP7 (R&D Systems) was produced in mouse myeloma cells. TGFβ neutralizing antibody (1D11, ThermoFisher) was used to exclude involvement of TGFβ contamination in CTHRC1-induced effects. To knockdown candidate genes, 50 nM siRNA pools (Microsynth) were delivered into mature adipocytes on day 13 using Lipofectamine RNAiMAX (Invitrogen) according to manufacturer's instructions. A control non-targeting siRNA pool was used in all experiments as a control. All siRNA sequences are listed in Supplementary Tab. 2. To overexpress GPR180, a lentivirus was generated (see below) and the cells were infected at day 11. After 24 h, transfection medium was replaced by fresh adipogenic medium. Adipocytes were cultured until day 17 or 18, when cellular respiration was determined, or cells were harvested for RNA and protein analysis. TGFβ in culture supernatants was measured by ELISA with sensitivity 8 pg/ml (ThermoFisher). All cell lines used were regularly tested negative for mycoplasma contamination throughout the whole duration of this study.

**Cell culture—immortalized murine brown adipocytes (iBAs)**. Preadipocytes isolated from the iBAT stromal-vascular fraction of late fetal and newborn C57Bl/6 mice (both genders) and immortalized by introducing the SV40 antigen were kindly provided by Prof. Klein (Klein et al., 2002). Preadipocytes (between passage 4 and 6) were grown on collagen-coated plates in DMEM containing 10% FBS and 1% Pen/Strep (Gibco) in normoxic humidified cell culture incubator (5% CO$_2$ and 37 °C). After reaching confluence, adipogenic differentiation was induced by supplementing the medium with IBMX (500 μM), dexamethasone (1 μM), insulin (20 nM), T3 (1 nM) and indomethacin (125 μM). All compounds were obtained from Sigma-Aldrich. After 48 h, medium was replaced by fresh maintenance medium containing (insulin and T3), which was replaced every other day. For siRNA-mediated knockdown, differentiating adipocytes (day 5) were trypsinized, counted and replated on collagen-coated multi-well plates to reduce cell density. Cells were allowed to attach, recover and maturate before siRNA transfection (100 nM siRNA pools, day 6). Cells were harvested on day 9 for RNA and protein analysis, or cellular respiration measurements. All siRNA sequences are listed in Supplementary Tab. 2. All cell lines used were regularly tested negative for mycoplasma contamination throughout the whole duration of this study.

**Cell culture—HEK-293T cells**. HEK-293T cells (Thermo Fisher) were grown in DMEM supplemented with 10% FBS and 1% Pen/Strep (Gibco) normoxic humidified cell culture incubator (5% CO$_2$ and 37 °C). At confluence, the cells were washed and starved for 2 h in DMEM only followed by acute CTHRC1 or TGFβ1 stimulus for signalling studies.

**Generation of stable HEK-293T cell line lacking GPR180**. Six oligo duplexes targeting human GPR180 gene (Supplementary Tab. 3) were synthesized by Microsynth Inc. and inserted into plasmid lentiCRISPRv2 (Addgene, #52961) via BsmBI restriction site. LentiCRISPRv2-Gpr180 gRNA plasmids were transfected into 293LTV cell line (Cell Biolabs) together with pMD2.G (Addgene, #12259) and psPAX2 (Addgene, #12260) by PEI in Opti-MEM medium. The virus-containing medium was collected 2 days later and concentrated using PEG-it Virus Precipitation Solution (SBI, LV825A-1) according to the manufacturer's instructions. The concentrated lentiviruses were added into medium to infect HEK-293T cells in presence of polybrene (Sigma, H9268). Two days later, the cells were suspended and selected by 1 μg/ml puromycin for 1 week to obtain a stable cell line. Afterwards, the HEK-293T cells were sorted into single cells using the Sony SH800S Cell Sorter (Sony Biotechnology) and cultured in 96-well plates for clonal expansion. Individual clones were validated using PCR and Sanger sequencing. Using this approach, we obtained a stable HEK-293T cell line with deletion of 83 bp (the sequence is available upon request) that led to a premature stop codon in exon 1 of GPR180 gene.

**Intraperitoneal glucose tolerance test**. To measure glucose tolerance, mice were fasted for 6 h by transfer to a clean cage without food and the test was performed at the end of the dark (active) phase. Mice were weighed and after fasting glucose levels were obtained from a tail vein using a standard glucometer (ACCU-CHEK Aviva, Roche), D-glucose (Sigma-Aldrich) was injected intraperitoneally at dose 1 mg/g body weight. Blood glucose levels were measured 15, 30, 60 and 120 m after glucose injection using glucometer.

**Body composition measurement**. Live mice body composition was measured with a magnetic resonance imaging technique (EchoMRI 130, Echo Medical Systems). Mice were fasted for 6 h before measurement. Fat and lean mass was analyzed using EchoMRI 14 software.

**Indirect calorimetry**. Indirect calorimetry measurements were performed with the Phenomaster (TSE Systems) using TSE PhenoMaster software v5.6.5 or Promethion (Sable Systems) according to the manufacturer's guidelines. O$_2$ and CO$_2$ levels were measured for 60 s every 20 m continuously. In case of GPR180 global knockout mice, following basal measurement mice were injected i.p. with CL-316.243 (0.1 mg/kg/day) to activate non-shivering thermogenesis. In case of adipocyte-specific GPR180 knockout animals, recombination of floxed allele was induced by intraperitoneal injection of 2 mg/mouse tamoxifen on 2 consecutive days after basal measurement. Regarding experiment with Cthrc1 overexpression, AAV injection was performed prior to housing animals in metabolic cages. Energy expenditure was calculated according to the manufacturer's guidelines. The respiratory quotient was estimated by calculating the ratio of CO$_2$ production to O$_2$ consumption. Animals were single-caged and acclimated to the metabolic cage for 48 h prior metabolic recording. Locomotor activity, food and water intake were monitored throughout the whole measurement.

**Surface temperature measurement**. Surface temperature was recorded with an infrared camera at room temperature (E60;FLIR) and analyzed with FLIR Tools software (FLIR).

**Tissue harvest**. Animals were euthanized singly in carbon dioxide atmosphere. All tissues were carefully dissected, weighed and either snap frozen in liquid nitrogen until further processing or fixed in 4% paraformaldehyde for tissue histology. Popliteal lymph nodes were carefully removed from iWAT for all gene and protein expression analyses. For RNA and protein isolation, whole adipose tissue depot was homogenized.

**Blood parameters**. Blood was collected from mice fasted for 6 h by cardiac puncture into EDTA coated tubes and plasma was obtained by centrifugation at 3000 r.p.m./20 min/4 °C. Plasma insulin was measured by Ultra-Sensitive Mouse Insulin ELISA kit (Crystal Chem), ALT activity by kinetic colorimetric assay (Sigma-Aldrich), free fatty acids by NEFA-HR(2) assay (Wako Chemicals), cholesterol by LabAssay Cholesterol (Wako Chemicals) and triglycerides by Cobas TRIGB kit (Roche/Hitachi), all following manufacturer instructions. Absorbance was measured by SynergyMx Plate reader and data analyzed by Gen5 v3.08 software (BioTek). CTHRC1 in human plasma was determined by sandwich ELISA following kit instructions (MMCRI)[55] We have validated the antibody by overexpression of CTHRC1 in HEK-293T cells, which do not express this protein, and determining CTHRC1 levels in culture media. Importantly, we also spiked recombinant CTHRC1 protein into plasma of patients with undetected circulating CTHRC1.

**Liver histology and lipid accumulation**. Part of liver left lobe was fixed in 4% paraformaldehyde for tissue histology. After 24 h, samples were transferred into 70% ethanol prior tissue processing. Tissues were dehydrated through graded alcohols, cleared with xylenes and infiltrated with paraffin by standard procedures. The blocks were sectioned at 4 μm cuts and stained with hematoxylin and eosin. Tissue sections were examined by light microscopy using AxioPhot microscope equipped with AxioCam MR (Zeiss). Another piece of left liver lobe was snap frozen in liquid nitrogen prior to lipid extraction. Total lipids were extracted using chloroform: methanol (2:1) mixture and normalized to tissue weight.

**Cellular respiration**. hMADS cells were differentiated on collagen-coated 96-well Seahorse microplates. On the day of experiment, adipogenic medium was replaced with XF Assay Medium (pH 7.4, Seahorse Bioscience) supplemented with glucose (1 g/L; Sigma-Aldrich), 2 mM sodium pyruvate (Invitrogen) and 2mM L-glutamine (Invitrogen). The oxygen consumption rate (OCR) was measured using the Extracellular flux analyzer XF96 and analyzed by Wave 2.6.0 (Agilent Seahorse). Test compounds were sequentially injected to obtain following concentrations: 1 μg/ml Oligomycin, 0.5 mM dibutyryl cAMP (1 μM isoproterenol for murine iBAs), 1 μg/ml FCCP, 3 μM Rotenone with 2 μg/ml Antimycin A. All compounds were purchased from Sigma-Aldrich, except for Oligomycin (Adipogen). OCR levels (pmol/min) were normalized to protein amount per well (μg protein). Non-mitochondrial respiration was subtracted to obtain basal, basal uncoupled, stimulated uncoupled and maximal mitochondrial respiration.

**Lentiviral GPR180 overexpression**. Lentiviral plasmid pLenti-MP2 was a gift from Pantelis Tsoulfas[63] and was modified by inserting IRES RFP cassette via XbaI and EcoRV restriction sites. Human GPR180 coding sequence (GenScript) was cloned under CMV promoter using XhoI and XbaI restriction sites using standard cloning techniques. GPR180 and control RFP lentiviruses were generated using pMDG2 and PAX2 packaging vectors in HEK-293T cells. Viral particles were precipitated by PEG-it Virus precipitation solution (BioCat). Biological titer was determined based on RFP positive HEK-293T cells transduced with virus. Mature hMADS-derived adipocytes were infected with 2 MOI with the aid of polybrene at final concentration of 8 μg/ml and Opti-MEM I reduced serum (Gibco).

**GPR180 topology**. To address GPR180 orientation in plasma membrane, lentiviral overexpression of the target´s coding sequence with epitope tags at different position was performed in mature adipocytes. HA tag was inserted between sequences corresponding to signal peptide and mature protein. V5 tag was inserted at carboxy terminus in front of stop codon. White adipocytes overexpressing HA-GPR180, GPR180-V5 or RFP on day 18 were washed with PBS and fixed 10 m in 4% Formaldehyde at 4 °C. After blocking with 5% BSA in PBS under permeabilizing (0.05% Triton X-100) or intact (no triton) conditions for 90 min RT, epitope tags were stained overnight at 4 °C with primary antibodies anti-HA (Cell Signaling) or anti-V5 (Invitrogen) diluted 1:500 in PBS. After cells were washed 3 times with PBS, secondary Alexa 488 antibody (ThermoFisher) diluted 1:500 in PBS was added for 1 h RT and nuclei were stained in parallel using Hoechst (Cell Signaling). After cells were washed 3 times with PBS, pictures were obtained using the automated Operetta imaging system (PerkinElmer). Cells overexpressing RFP were used to subtract background of unspecific antibody binding.

**Assessment of SMAD3 shuttling by immunofluorescence**. To address SMAD3 shuttling in response to CTHRC1, mature beige adipocytes at day 18 were fasted for 2 h and then acutely stimulated with recombinant CTHRC1 for 1 h. TGFβ was used as positive control. Then, cells were washed with PBS and fixed 20 m with 4% Formaldehyde at RT. After washing with PBS, cells were incubated with 5% acetic acid in ethanol for 20 min at −20 °C to remove lipids. After washing with PBS and subsequent blocking with 5% BSA in PBS under permeabilizing (0.05% Triton X-100) conditions for 90 min RT, SMAD3 was stained overnight at 4 °C with primary antibodies either anti-SMAD3 (Cell Signaling) or phosphor-SMAD3$^{Ser423}$ (Abcam) diluted 1:500 in blocking buffer. After cells were washed 3 times with PBS, secondary anti-rabbit Alexa 488 antibody (ThermoFisher) diluted 1:500 in blocking buffer was added for 1 h RT and nuclei were stained in parallel using Hoechst (Cell Signaling). After cells were washed 3 times with PBS, pictures were obtained using the automated Operetta imaging system (PerkinElmer). The positive pSMAD3 cells were quantified based on standard image processing steps including thresholding, size filtering (for the nuclei) and by counting those cells whose nuclei overlap with the pSMAD3 staining. For the quantification of cytosolic/nucleus ratio, around each segmented nucleus at least 3 pixel wide ring was created by binary operations as accurate estimation of the individual cell borders were not possible. The ratio is determined by the average pSMAD3 signal intensity inside the nucleus relative to the average signal intensity within the ring. The quantified data has been filtered according to circularity of the nucleus (0.5 <), estimated background (weak) signal for the pSMAD3 staining, and nucleus/cytosolic ratio to hinder the artefactual influence of the segmentation (border estimation of the nuclei). The image processing is performed with Matlab 2019a.

**Signalling studies**. Phosphorylation of SMAD3 in beige adipocytes in response to TGFβ1 following knockdown of GPR180 was determined by SMAD3 (pSer423/S425) ELISA kit (Abcam) while silencing of TGFβR2 was used as negative control. Involvement of Gq signalling following GPR180 knockdown or in response to CTHRC1 was assessed by fluorescent calcium indicator Fluo-4 (ThermoFisher) or IP1 ELISA (Cisbio) following manufacturer instructions. Briefly, mature beige adipocytes were starved 2 h prior calcium measurement. Then the cells were loaded with cell permeable Fluo-4 and probenecid mixture and incubated 45 min at 37 °C. After washing, the cells were incubated in buffer (130 mM NaCl, 5 mM KCl, 10 mM HEPES, 2 mM CaCl$_2$, 10 mM glucose, pH 7.4) for 20 min and fluorescence was measured with excitation at 494 nm and emission at 516 nm. After basal measurement, CTHRC1 or control were added and kinetic measurement continued. In the end of the assay, 2.5 μM ionophore A23187 was added as positive control to validate the assay. Fluorescence was normalized to nuclei number stained by Hoechst as described above. In case of IP1 measurement, 2 h of starvation of beige adipocytes was followed by CTHRC1 stimulus for 15 min in the presence of lithium chloride to suppress IP1 degradation. Endothelin (0.3 nM) was used as positive control. Absorbance was normalized to protein content. Levels of cAMP following GPR180 knockdown or in response to CTHRC1 were determined by Direct cAMP ELISA kit (Enzo) in basal or forskolin (10 μM) pre-treated adipocytes for 15 min and normalized to protein content.

**Binding studies**. A HiBiT tag was cloned at the C terminus of human CTHRC1 coding sequence and inserted into pcDNA3.1 expression vector (Invitrogen) to obtain a HiBiT-tagged CTHRC1. Three days after delivery of the overexpression construct into HEK-293T cells, standard culture medium (DMEM + 10% FBS + 1%Pen/Strep) was replaced by DMEM + 0.5% and collected after 8 h. Medium containing HiBiT-tagged CTHRC1 was centrifuged 300 × g for 5 m and sterile filtered. Wild-type and GPR180$^{−/−}$ HEK-293T cells were grown on 15 cm dishes until confluence, washed with PBS and serum starved in 15 ml DMEM for 2 h prior to the binding assay. One milllilitre of medium containing HiBiT-tagged CTHRC1 protein was added to the starvation medium after 2 h and incubated for 10 m at 37 °C. Cells were washed 5 times with ice-cold PBS to wash away unbound CTHRC1 and frozen at −80 °C until protein extraction. Interaction of CTHRC1 with GPR180 was studied using the Nano-Glo® HiBiT Blotting System (Promega) according to manufacturer's instructions.

**RNA extraction, cDNA synthesis, quantitative RT-PCR**. Total RNA was extracted from tissues or cells using Trizol reagent (Invitrogen) according to the manufacturer's instructions. DNase treatment (NEB BioLabs) was included to remove traces of genomic DNA. Reverse transcription was performed to generate cDNA library by using the High Capacity cDNA Reverse transcription kit (Applied Biosystems), with 1 μg of RNA. Quantitative PCR was performed on a ViiA7 (Applied Biosystems) and relative mRNA concentrations normalized to the expression of RPL13A1 (hMADS), B2M (human adipose tissue) or TBP (iBAs and adipose tissue) were calculated by the ΔΔCt method using ViiA7 Ruo v1.2.3 software Primer sequences are found in Supplementary Tab. 4.

**Protein extraction and western blot**. Adipose tissue samples and in vitro differentiated adipocytes were homogenized in RIPA buffer (50 mM Tris-HCl pH 7.4, 150 mM NaCl, 2 mM EDTA, 1.0% Triton X-100, 0.5% sodium deoxycholate) supplemented with protease (Complete, Roche) and phosphatase (Halt phosphatase inhibitor cocktail, ThermoFisher) inhibitor cocktails. Lysates were cleared by centrifugation at 12,000 × g for 15 m at 4 °C. Protein concentration of the supernatants was determined by DC Protein Assay (Bio-Rad). Equal amount of proteins (5–20 μg) were separated on 12% SDS-polyacrylamide gel, transferred to a nitrocellulose membrane (Bio-Rad) and stained for UCP1 (1:1000, ThermoFisher), phospho-SMAD3 (Ser423/425; 1:1000, Abcam), phospho-SMAD1/5/9 (Ser463/465; 1:1000, Abcam), SMAD3 (1:3000, Abcam), CTHRC1 (1:750, Sigma–Aldrich), TGFβ 1,2,3 (1:1000, R and D Systems), TGFβR1 (1:2000, Abcam), TGFβR2 (1:1000, Abcam), RFP (1:2000, Evrogen), OXPHOS (1:500, Abcam), phospho-p44/42 MAPK (ERK1/2) (Thr202/Tyr204; 1:1000, Cell Signaling), p44/42 MAPK (ERK1/2) (1:1000, Cell Signaling), phospho-p38MAPK (Thr180/Tyr182; 1:1000, Cell Signaling), p38MAPK (1:1000, Cell Signaling), phospho-HSL (Ser660; 1:1000, Cell Signaling), phospho-PKA substrates (1:1000, Cell Signaling), phospho-AMPK (Thr172; 1:1000, Cell Signaling), phospho-CREB (Ser133; 1:1000, Cell Signaling), phosphor-FAK (Tyr397; 1:1000, Cell Signaling), phospho-FOXO1 (Thr24; 1:1000, Cell Signaling), phospho-JNK (Thr183; 1:1000, Cell Signaling), phospho-AKT (Thr308; 1:1000, Cell Signaling), HSP90 (1:1000, Cell Signaling) and γ-tubulin (1:10.000, Sigma–Aldrich). Signal of the HRP-conjugated secondary antibodies (1:10.000, Merck) was visualized by the Image Quant system (GE Healthcare Life Sciences). Quantification of western blots was done using ImageJ version 1.53e (NIH).

**Analysis of adipocyte differentiation**. Mature beige adipocytes at day 18 after GPR180 silencing were used for differentiation analysis. Briefly, cells in 96-well optical plate were fixed with 4% formaldehyde for 20 min and washed 3 times with PBS. Immediately after washing, cells were stained with Bodipy (Invitrogen) for lipid droplets and Hoechst (Cell Signaling) for nuclei. Twenty-five pictures per well were taken with an automated microscope imaging system (Operetta, PerkinElmer). Pictures were analyzed using the Harmony software v3.5. In differentiation assay, all cells (Hoechst stained nuclei) surrounded by lipid droplets were considered adipocytes.

**SMAD reporter assay**. Naïve and GPR180 knockout HEK-293T cells at 80% confluence were transfected with 500 ng/ml either control pBV luciferase reporter plasmid with very low basal activity or SBE4-Luc luciferase reporter containing four copies of SMAD binding element utilizing Turbofect transfection reagent (Thermo Scientific). Four hours after transfection, medium was changed to DMEM supplemented with 0.5% FBS together with either control, CTHRC1 or TGFβ1 treatment. After overnight incubation (18 h), the cells were lysed in passive lysis buffer and luminescence was measured by Dual Luciferase Assay kit (Promega) according manufacturer's instruction. Firefly luminescence of the reporter was normalized to renilla luminescence that was co-transfected together with reporter plasmids as internal control. In addition, SBE4 luciferase activity was normalized to the luminescence signal of control pBV luciferase plasmid with corresponding treatment. Plasmids pBV-Luc and SBE4-Luc were a gift from Bert Vogelstein[64].

**RNA sequencing, mapping and analysis**. RNA extracted from brown and white adipose tissue biopsies was quality checked by Tapestation (Agilent). All samples had a RIN value of greater than 8. For the preparation of libraries the TruSeq mRNA sample preparation kit (Illumina) was used. Sequencing was performed as 50 bp, single reads and 7 bases index read on an Illumina HiSeq2000 instrument. Approximately 20–30 million reads per sample were obtained.

RNA sequencing data were processed utilizing kallisto[65] with the gencode human reference genome release 29[66]. The different datasets were processed by the same pipeline and reference genome to make them comparable. The quality of the fastq files was checked by the FastQC program[67]. The raw gene counts obtained from kallisto were processed by DESeq2[68] to call the DE genes between supraclavicular BAT over subcutaneous WAT and brown over white hMADS-derived adipocytes. The PCA plots after vst normalization of top 500 most variable genes were visually inspected to remove one outlier sample in case of supraclavicular BAT. DE genes with FDR cut-off <0.05 were selected and the overlapping DE genes in adipose tissue and hMADS were further processed. The list of membrane proteins was obtained from Uniprot (uniprot.org) using a custom query to filter out membrane proteins and the list of receptors was obtained from Baderlab website http://baderlab.org/CellCellInteractions. The overlapping DE

genes were filtered for membrane proteins and receptors. Furthermore, the genes with median expression of tpm <2 were filtered out. Lastly, the resulting genes were filtered for log2 fold change from the DE analysis.

We processed the GPR180 KD and control RNA-seq data in the same manner as described above using kallisto followed by DESeq2 and PCA analysis. In the DESeq2 workflow, first the count data was normalized by the median of ratios method. Next, the dispersion or biological variance was estimated. Thereafter, a generalized linear model was fitted for each gene to detect DE genes. We used the batch number as an additional covariate apart from condition in the generalized linear model of negative binomial distribution to account for the batch effect. The p-values obtained by Wald test were corrected by Benjamini–Hochberg multiple testing procedure. The DE genes were analyzed for KEGG pathway enrichment using GSEA pre-ranked method, which enables the analysis of up and downregulated genes simultaneously. This approach significantly improved the sensitivity of the geneset enrichment analysis. The genes were sorted by their log2 fold change and analyzed by the GSEA pre-ranked algorithm implemented in GSEApy (https://github.com/zqfang/GSEApy). The data were processed by in house scripts written in R (r-project.org) with tidyverse libraries and python (python.org) with pandas library following standard DeSeq2 and GSEA tutorials.

**Proteomic analysis of cell-conditioned media.** For identification of proteins secreted by hMADS cells, we collected 10 ml of cell-conditioned media. The medium was filtered (0.45 μm syringe filter) to remove cellular debris and concentrated 10 times using Amicon Ultra-15 centrifugal filter units with a 3 kDa cut-off (Millipore). For western blot, 30 μl of concentrated medium was loaded into SDS-PAGE gel to detect TGFβ isoforms or CTHRC1. For mass-spectrometry analysis, proteins and peptides present in the concentrated medium were pre-cipitated using ice-cold 10% TCA. Protein pellets were washed twice with ice-cold acetone and re-solubilized in 90 μl of 10 mM Tris, 2 mM $CaCl_2$, pH 8.2. Proteins were digested with 10 μl sequencing grade Trypsin (Promega; 100 ng/μl in 10 mM HCl) and 10 ul RapiGest (1% in water) for 16 h at 37 °C. After tryptic digestion, samples were briefly spun; supernatants were dried, dissolved in 20 μl 0.1% formic acid and transferred to autosampler vials for LC/MS/MS. Database search was performed by using the Mascot (SwissProt). We filtered out the overlapping genes from MS and hMADS DE genes in brown over white adipocytes. We obtained a list of secreted proteins from Uniprot. The list of extracellular matrix components or proteins regulating immune response were obtained by GO and Reactome anno-tations and was used to narrow down the list of secreted proteins.

**Quantification and statistical analysis.** For in vivo studies, littermates were used for all experiments. Sample sizes were determined on the basis of previous experiments using similar methodologies. The animal numbers used for all experiments are indicated in the corresponding figure legends. All animals were included in statistical analyses, and the investigators were not blinded. All cell culture experiments were performed with 2–3 technical replicates for RNA and protein analysis, 5–6 replicates for measurement of cellular respiration, and independently reproduced 2-4 times. Results are reported as mean ± SEM for mouse and cell culture data. Two-tailed unpaired Student's t-test was applied on comparison of two groups and one-way ANOVA with Dunnett's post-hoc test was applied on comparisons of multiple groups. Paired Student's t-test was used to analyse the differences in paired samples (e.g. BAT and WAT from the same patient). When effect of two factors (e.g. knockdown and treatment) was investigated, two-way ANOVA with Sidak post-hoc test was performed. Parameters measured over the time (e.g. Phenomaster data) were analyzed by two-way ANOVA with repeated measurements followed by Sidak post-hoc test. Pearson's correlation coefficient was calculated to address association of two parameters. Fisher's exact test was used to analyse contingency table (detectability of circulating CTHRC1 in human study with different patient groups). All statistical analyses were performed using GraphPad Prism 8. Statistical differences are indicated as * for $P < 0.05$, ** for $P < 0.01$ and *** for $P < 0.001$.

**Reporting summary.** Further information on research design is available in the Nature Research Reporting Summary linked to this article.

## Data availability
The RNA sequencing data of hMADS cells after ablation of GPR180 denerated in this study have been deposited in The European Nucleotide Archive under accession number PRJEB38756. The RNA sequencing data from the clinical transcriptome study used in this study are available in The European Nucleotide Archive,accession number PRJEB23275. The mass-spectrometry proteomics data have been deposited to the ProteomeXchange Consortium via the PRIDE[69] partner repository with the dataset identifier PXD029335. Supporting data are included in this article as supplementary data 1-4. Source data are provided with this paper.

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

## Acknowledgements

We thank to Christoph Reuss and Christofer Tautermann from Boehringer Ingelheim for their valuable input and technical support. Proteomic analysis was performed by Functional Genomics Center Zurich. The work was supported by The EFSD New Horizons grant (J.U., C.W.); Scientific Grant Agency of SAS VEGA 2/0096/17 (J.U.). The schematic overview of GPR180 mechanism in Fig. 7i was created by modification of elements from Servier Medical Art (smart.servier.com).

## Author contributions

L.B., H.N. and C.W. designed the study; L.B., M.B., B.H. and C.W. supervised the experiments; L.B., M.B., C.H., C.M., E.K., W.S., H.D., L.D. and V.E. performed the experiments; C.H. and A.H. performed image analysis; P.N., K.V. and T.N. performed the clinical transcriptome study, A.G. and U.G. performed all bioinformatics analyses; M.B., B.U., J.U. and Z.K. performed the clinical study and analyzed human samples; E.A. provided resources; P.P. generated global knockout; T.L. generated AAV; L.B., M.B. and C.W. wrote the paper; all authors reviewed and edited the manuscript.

## Competing interests

The authors declare no competing interests.
