## [Peer Review File · Nature Communications]

GPR180 is a component of TGF β signalling that promotes thermogenic adipocyte function and mediates the metabolic effects of the adipocyte-secreted factor CTHRC1REVIEWER COMMENTS

Reviewer #1 (Remarks to the Author):

NCOMMS-21-22395-T

Orphan GPR180 adopted: a novel component of TGF β family that promotes thermogenic adipocyte function by mediating effects of CTHRC1

This is a manuscript with experiments performed at a very high standard, conceptualized and executed by leaders in the field. The manuscript is well written, the rationale for the approach well-laid out, and the findings are novel.

I carefully read the comments of the other Reviewers. The authors went a long way to address all my comments and the points raised by the other two Reviewers. Moreover, as a pharmacologist, I do not agree with the comment about the magnitude of the effects shown in the ms, which are comparable to other studies recently published in high ranking scientific journals on the topic of thermogenic fat.

Reviewer #2 (Remarks to the Author):

In this revised version, the authors have made significant efforts to respond to the questions that were raised. They have mostly answered the questions with experiments and further discussion. I have three minor outstanding points that in my opinion would need to be addressed to support the overall impact and conclusions of the paper, but overall, I support the manuscript.

1. In the newly added metabolic chamber experiments, I believe that the statistical analyses are not performed correctly and the data are not corrected for multiple testing. This was also a problem with many of the older figures as well. Although the figure legend states 2-way Anova for repeated measures, individual p-values are displayed in the figures indicating that individual t-tests have been performed. If this is the case, Anova analyses need to be performed on the entire datasets. If the results are not statistically significant after recalculating the data, the conclusions are not supported. The figures in question are:

Fig 2c
Fig 2f
Fig 2g
Fig 2i
Fig 2n
Fig 2q
Fig 2r
Fig 6i
Fig 6j
Fig 6k

2. The authors have added cAMP and PKA data showing no changes when GPR180 is knocked down, in line with the statement that GPR180 is not a functional GPCR. However, I still have some concerns about the claim that CTHRC1 is a ligand for GPR180. First, no direct cell-free ligand-receptor binding studies have been performed in the revised version to show that CTHRC1 binds GPR180. The HiBiT assay lacks positive and negative controls to definitely determine if the binding is truly specific. Based on the presented data and if no new data will be added, there is not sufficient support for the claim of a "novel ligand-receptor pair" and such statements should be rephrased to state only what is shown, which is that GPR180 is required for some actions of CTHRC1.

3. The concern with TGF-beta as a contaminant in the recombinant CTHRC1 protein preparation could be better addressed in my opinion. The authors have tried to address the issue by performing the experiments in the presence of an TGF-beta antibody to neutralize a possible contaminant. A better approach would be to actually measure the levels of all proteins (including TGF-beta) by mass spectrometry in the recombinant protein preparation that is being used. This is

a simple experiment that would definitively rule out any issues with protein contaminants and should be a minimal requirement for any experiment that uses recombinant proteins.

Reviewer #3 (Remarks to the Author):

The authors have successfully addressed my previous concerns, which centered around gaining greater insight into the manner by which GPR180 and CTHRC1 modulate TGFb1/SMAD3 signaling, and characterizing the biophysical interaction between GPR180 and CTHRC1. The additional experiments and explanation are greatly appreciated and have added much clarity to the manuscript. The careful framing and wording of the results, and toning down the previous claim that GPR180 represents a novel TGFb1 receptor are appropriate given the limits of the current data. The authors have made a very sincere effort to address all of the detailed concerns from me and the other reviewers and should be commended for the thoroughness of the revision.

REVIEWER COMMENTS

Reviewer #1 (Remarks to the Author):

NCOMMS-21-22395-T

Orphan GPR180 adopted: a novel component of TGF β family that promotes thermogenic adipocyte function by mediating effects of CTHRC1.

This is a manuscript with experiments performed at a very high standard, conceptualized and executed by leaders in the field. The manuscript is well written, the rationale for the approach well-laid out, and the findings are novel.

I carefully read the comments of the other Reviewers. The authors went a long way to address all my comments and the points raised by the other two Reviewers. Moreover, as a pharmacologist, I do not agree with the comment about the magnitude of the effects shown in the ms, which are comparable to other studies recently published in high ranking scientific journals on the topic of thermogenic fat.

We thank Reviewer #1 for thorough revision of our manuscript and the positive feedback.

Reviewer #2 (Remarks to the Author):

In this revised version, the authors have made significant efforts to respond to the questions that were raised. They have mostly answered the questions with experiments and further discussion. I have three minor outstanding points that in my opinion would need to be addressed to support the overall impact and conclusions of the paper, but overall, I support the manuscript.

We thank Reviewer #2 for thorough revision and the suggestions, which helped to improve the quality of our manuscript.

1. In the newly added metabolic chamber experiments, I believe that the statistical analyses are not performed correctly and the data are not corrected for multiple testing. This was also a problem with many of the older figures as well. Although the figure legend states 2-way Anova for repeated measures, individual p-values are displayed in the figures indicating that individual t-tests have been performed. If this is the case, Anova analyses need to be performed on the entire datasets. If the results are not statistically significant after recalculating the data, the conclusions are not supported. The figures in question are:

We thank Reviewer for bringing up this point. With respect to old figures and displaying individual p-values, we indicated significance in figures based on post-hoc (multiple comparison) tests which are integral part of ANOVA. We also re-analyzed new metabolic chamber experiments as suggested by the reviewer. To avoid any misunderstanding, we provide here a detailed explanation for each figure.

Fig 2c

Comparison of the area under the curve that reflects cumulative energy expenditure over the time course of measurement revealed significantly lower energy expenditure in *Gpr180* knockout mice ($p < 0.001$). Analyses of the entire dataset by ANOVA with repeated measures revealed a strong trend towards a decrease in energy expenditure in global *Gpr180* knockout mice $F(1,10)=3.764$, $p = 0.0811$. We have added this to the revised paper.

Fig 2f

A 2-way ANOVA with repeated measures was used to analyse differences between the genotypes during the timecourse of glucose tolerance test with following results: (source of variation and significance) factor genotype $F(1, 11) = 18.40$, $p = 0.0013$; factor time $F(4, 44) = 141.0$, $p < 0.001$; and interaction between genotype and time $F(4, 44) = 6.214$, $p = 0.0005$ with subsequent Sidak multiple comparison test that revealed significant differences between genotypes at individual timepoints as indicated in the graph.

Fig 2g

A 2-way ANOVA with repeated measures revealed significant differences in body weight between the genotypes in response to high fat diet feeding regime with source of variation and significance as follows: factor genotype $F(1, 25) = 13.10$, $p = 0.0013$; factor time $F(12, 300) = 393.0$, $p < 0.001$ and interaction between factors genotype and time $F(12, 300) = 5.712$, $p < 0.001$. Subsequent Sidak multiple comparison test revealed significant differences between genotypes at individual timepoints as indicated in the graph.

Fig 2i

A 2-way ANOVA with repeated measures was used to analyse differences between the genotypes during the timecourse of glucose tolerance test with following results: (source of variation and significance): factor genotype $F(1, 10) = 5.212$, $p = 0.0456$ and factor time $F(4, 40) = 175.6$, $p < 0.001$. As before, no interaction between factors genotype and time $F(4, 40) = 0.9283$, $p = 0.4573$ was found. Similarly, Sidak multiple comparison test did not reveal any significant differences between the genotypes at individual timepoints, thus, no statistical significance is indicated in the graph.

Fig 2n

Analysis of the entire dataset of energy expenditure measurement in adipocyte specific *Gpr180* knockout mice by ANOVA with repeated measures revealed a significant interaction between the factors time and genotype ($F(83, 734) = 1.791$, $p < 0.001$) as expected based on the experimental design with knockout induced after baseline measurement. Subsequent Fisher's LSD multiple comparison test revealed significant differences between the genotypes at individual timepoints as indicated in the graph. Data was not corrected for multiple comparison in line with planned comparison with respect to the experimental design.

Fig 2q

A 2-way ANOVA with repeated measures revealed a strong trend towards impaired glucose utilization following *Gpr180* ablation in adipocytes (factor genotype) $F(1, 6) = 4.912$, $p = 0.0685$ during timecourse of glucose tolerance test and significant difference for factor time $F(4, 24) = 130.4$, $p < 0.001$; as well as significant interaction between factors genotype and time $F(4, 24) = 7.952$, $p = 0.0003$ with subsequent Sidak multiple comparison test that revealed significant differences between the genotypes at individual timepoints as indicated in the graph.

Fig 2r

A 2-way ANOVA with repeated measures revealed significant differences in body weight between genotypes in response to high fat diet feeding regime with source of variation and significance as follows: factor genotype $F(1, 11) = 5.646$, $p = 0.0367$; factor time $F(10, 110) = 186.0$, $p < 0.001$ and

interaction between factors genotype and time $F(10, 110) = 4.529$, $p < 0.001$. Subsequent Sidak multiple comparison test revealed significant differences between the genotypes at individual timepoints as indicated in the graph.

Fig 6i

Comparison of area under the curve that reflects cumulative energy expenditure over time of measurement revealed significantly higher energy expenditure in CTHRC1 overexpressing mice ($p = 0.0014$). Multiple t-test revealed several significant timepoints as indicated in the graph. Analyses of the entire dataset by a 2-way ANOVA with repeated measures did not reveal significant differences in energy expenditure following administration of CTHRC1 aav ($F(1,12)=1.895$, $p = 0.1938$).

Fig 6j

Analyses of the entire dataset by ANOVA with repeated measures revealed significant differences in respiratory exchange ratio upon aav treatment $F(1, 12) = 22.25$, $p = 0.0005$. In addition, significant interaction between the factors aav and time $F(95, 1140) = 1.914$, $p < 0.001$ was found. Subsequent Sidak multiple comparison test revealed significant differences between the genotypes at individual timepoints as indicated in the graph.

Fig 6k

A 2-way ANOVA with repeated measures was used to analyse differences between experimental groups as indicated in the graph legend during the timecourse of glucose tolerance test with following results: (source of variation and significance) factor experimental group $F(3, 31) = 21.06$, $p < 0.001$; factor time $F(15, 155) = 206.4$, $p < 0.001$; and interaction between experimental group and time $F(15, 155) = 8.711$, $p < 0.001$ with subsequent Sidak multiple comparison test that revealed significant differences between experimental groups at individual timepoints as indicated in the graph.

2. The authors have added cAMP and PKA data showing no changes when GPR180 is knocked down, in line with the statement that GPR180 is not a functional GPCR. However, I still have some concerns about the claim that CTHRC1 is a ligand for GPR180. First, no direct cell-free ligand-receptor binding studies have been performed in the revised version to show that CTHRC1 binds GPR180. The HiBiT assay lacks positive and negative controls to definitely determine if the binding is truly specific. Based on the presented data and if no new data will be added, there is not sufficient support for the claim of a "novel ligand-receptor pair" and such statements should be rephrased to state only what is shown, which is that GPR180 is required for some actions of CTHRC1.

We thank Reviewer for the comment. We rephrased the manuscript according to the reviewers suggestions.

3. The concern with TGF-beta as a contaminant in the recombinant CTHRC1 protein preparation could be better addressed in my opinion. The authors have tried to address the issue by performing the experiments in the presence of an TGF-beta antibody to neutralize a possible contaminant. A better approach would be to actually measure the levels of all proteins (including TGF-beta) by mass spectrometry in the recombinant protein preparation that is being used. This is a simple experiment

that would definitively rule out any issues with protein contaminants and should be a minimal requirement for any experiment that uses recombinant proteins.

We addressed contamination of recombinant CTHRC1 protein by TGF β as suggested by Reviewer 3, who raised the comment during the first round of revision. We could clearly reproduce all CTHRC1 effects in the presence of a high dose of TGF β neutralizing antibody. To address the additional concerns of Reviewer 2, we submitted both CTHRC1 recombinant proteins from different vendors that were used in our study to Mass Spectrometry analysis, as suggested. In both recombinant proteins of different origin, CTHRC1 was the most abundant protein. As expected, several other proteins were identified to be present in the samples out of which most represented structural protein, intracellular enzyme or heat shock protein contaminants. We identified 16 overlapping contaminants between the two recombinant proteins (please see the list below) which are minor components and not relevant to TGF β signalling. Importantly, none of the recombinant proteins was contaminated by TGF β . As we could reproduce phenotypes with two recombinant CTHRC1 proteins of different origins, we are confident that the observed phenotype is attributed to CTHRC1. In addition we would like to point out that the increase in UCP1 expression and/or SMAD3 phosphorylation upon Cthrc1 treatment is dependent on GPR180 and lost upon GPR180 knockdown.

Gene Name	Identified Protein
HSPA1A	Cluster of Heat shock 70 kDa protein 1A
TUBA1B	Cluster of Tubulin alpha-1B chain
ACTB	Cluster of Actin, cytoplasmic
CFI	Cluster of Complement factor I
HSPA8	Heat shock cognate 71 kDa protein
EEF1A1	Cluster of Elongation factor 1-alpha 1
PLOD3	Multifunctional procollagen lysine hydroxylase and glycosyltransferase
PLOD1	Procollagen-lysine,2-oxoglutarate 5-dioxygenase 1
YWHAE	Cluster of 14-3-3 protein epsilon
RYDEN	Repressor of yield of DENV protein
HSPA5	Endoplasmic reticulum chaperone BiP
IGHG1	Immunoglobulin heavy constant gamma 1
NME2	Cluster of Nucleoside diphosphate kinase B
PRDX4	Peroxiredoxin-4
STC2	Stanniocalcin-2
TXN	Thioredoxin

Reviewer #3 (Remarks to the Author):

The authors have successfully addressed my previous concerns, which centered around gaining greater insight into the manner by which GPR180 and CTHRC1 modulate TGF β 1/SMAD3 signaling, and characterizing the biophysical interaction between GPR180 and CTHRC1. The additional experiments and explanation are greatly appreciated and have added much clarity to the manuscript. The careful framing and wording of the results, and toning down the previous claim that GPR180 represents a novel TGF β 1 receptor are appropriate given the limits of the current data. The authors have made a very sincere effort to address all of the detailed concerns from me and the other reviewers and should be commended for the thoroughness of the revision.

We thank Reviewer #3 for thorough revision and positive feedback.

REVIEWERS' COMMENTS

Reviewer #2 (Remarks to the Author):

The authors have thoroughly revised the manuscript and even added new data in a third revision. I congratulate the authors for publishing this interesting study.